# Impact of Pacific Ocean heatwaves on phytoplankton community composition

Lionel A. Arteaga [1,2] ✉ & Cecile S. Rousseaux [3]

Since 2013, marine heatwaves have become recurrent throughout the equatorial and northeastern Pacific Ocean and are expected to increase in intensity relative to historic norms. Among the ecological ramifications associated with these high temperature anomalies are increased mortality of higher trophic organisms such as marine mammals and seabirds, which are likely triggered by changes in the composition of phytoplankton, the base of the marine trophic food web. Here, we assimilated satellite ocean color data into an ocean biogeochemical model to describe changes in the abundance of phytoplankton functional types (PFTs) during the last decade's (2010s) warm anomalies in the equatorial and northeastern Pacific Ocean. We find important changes associated with the "Blob" warm anomaly in the Gulf of Alaska, where reduced silica supply led to a switch in community composition from diatoms to dinoflagellates, resulting in an increase in surface ocean chlorophyll during the Summer–Fall of 2014. A more dramatic change was observed in the equatorial Pacific, where the extreme warm conditions of the 2016 El Niño resulted in a major decline of about 40% in surface chlorophyll, which was associated with a nearly total collapse in diatoms.

[1] Global Modeling and Assimilation Office, NASA Goddard Space Flight Center, Greenbelt, MD, USA. [2] Goddard Earth Sciences, Technology and Research II, University of Maryland Baltimore County, Baltimore, MD, USA. [3] Ocean Ecology Laboratory, NASA Goddard Space Flight Center, Greenbelt, MD, USA. ✉email: lionel.arteagaquintero@nasa.gov

In the fall of 2013, a persistent warm temperature anomaly (referred to as "The Blob") began to develop in the northeast Pacific Ocean, initially spreading over the Gulf of Alaska and then extending southwards over most of the Pacific coast of North America by the end of 2014[1,2]. High sea surface temperature (SST) anomalies continued to be recorded in the equatorial Pacific with the onset of El Niño conditions during the fall of 2014, which developed into one of the most extreme El Niño events ever recorded, reaching warm SST anomalies close to 3 °C by 2015–2016[3,4]. Oceanic heat anomalies reintensified again in the northeast Pacific with a resurgence of a Blob-like anomalous SST pattern in 2019[5], making the 2010s a decade of frequent marine heatwaves in the Pacific Ocean, consistent with ongoing background warming of the surface ocean associated with anthropogenic forcing[6,7].

The Pacific Blob has been associated with numerous deleterious environmental changes observed in the Gulf of Alaska and along the west coast of North America between 2013 and 2016. Changes at the highest trophic levels included mass die-offs (approximately 1 million) of seabird[8], increased whale mortality[9], decreased recruitment and increased mortality in Alaskan salmon and pollock[10], as well as a stock collapse in Pacific cod and forage fish communities[11,12]. As the warm anomaly expanded southwards throughout the California Current System (CCS), sardine and anchovy were mostly absent, whereas their larvae were observed in colder waters further north off the coast of Oregon and Washington for the first time in many years[13]. Many of the changes observed in the population of fish, marine mammals, and seabirds are linked to alterations in the zooplankton community and lower nutritional value of taxa related to warm waters[14]. Zooplankton taxa from subtropical warm areas tend to be smaller in size and have less lipid content (associated with overwintering strategies in subarctic populations), thus providing fewer calories to predators[10,13,14]. In spring of 2015 subarctic copepods were rare along the coast of Oregon, while warm species proliferated[13]. Also recorded that year was one of the lowest abundances of Krill (*Euphausia pacifica*) in 18 years of monitoring in the southern CCS[15].

At the core of these ecological alterations in the marine food web is a likely decline in the biomass of primary producers (i.e., phytoplankton) and/or a change in their community composition. Photosynthesis by phytoplankton represents the gateway of biochemical energy in the form of organic carbon into the marine ecosystem, thereby directly (or indirectly) sustaining all other heterotrophic organisms in the surface and mesopelagic ocean. The assimilation and export of organic carbon towards greater depths mediated by phytoplankton also plays a key role in regulating atmospheric carbon dioxide ($CO_2$) levels[16]. Satellite-based estimates indicate that in 2014 the winter chlorophyll (Chl) concentration in the highly productive transition zone between subarctic and subtropical waters of the northeast Pacific Ocean dropped to their lowest winter level since 1997[17]. Widespread changes in phytoplankton community composition associated with the Pacific warm anomaly are considerably more challenging to detect. To date, the most impactful alteration identified occurred in early 2015 when record-breaking concentrations of the neurotoxin, domoic acid (DA), were initially detected in zooplankton net tow samples and subsequently in benthic populations of clams and mussels along the North American west coast affected by the expansion of the Blob.[18,19] The DA outbreak was created by a coastwide bloom of the toxigenic diatom *Pseudo-nitzschia*. Regional "hotspots" of toxic algae blooms are not uncommon along the U.S West Coast, but the 2015 coastwide toxic algal bloom was unprecedented and likely driven by the anomalous warm conditions[18].

Changes in the community partitioning of non-toxic phytoplankton groups with the potential to alter the transfer of energy

throughout the food chain or the downward export of carbon are less documented and constrained to narrow geographical areas or specific oceanographic stations[20–23]. In this study, we describe the most notorious rearrangements in phytoplankton functional types (PFTs) during the last decade's (2010s) warm anomalies in the northeastern and equatorial Pacific Ocean, inferred from a satellite ocean color data-assimilation framework. PFTs are conceptual assemblages of phytoplankton species with common ecological or biogeochemical roles in the marine ecosystem. Our analysis is primarily based on the assimilation of NASA's ocean color satellite record from the Moderate Resolution Imaging Spectroradiometer (MODIS) into an ocean biogeochemistry general circulation model (Methods). This assimilation framework has the advantage of overcoming spatial and temporal gaps in satellite retrievals and transforming remote sensing snapshots of a limited number of ocean-color variables into derived products such as biogeochemical fluxes and PFTs. We identified two periods where important changes in the partitioning of PFTs took place. The first one occurred between May and December of 2014 in the Gulf of Alaska where reduced nutrient (silicate) supply led to a decrease in diatoms, a silicifying phytoplankton associated with efficient carbon export and transfer of organic energy to higher trophic levels, permitting dinoflagellates to occupy a larger portion of the total stock in surface phytoplankton biomass. This switch in community composition prolonged an increase in surface chlorophyll concentration initiated since the beginning of the anomalous warm period in this region. The second event was detected during the 2016 El Niño (November 2015–March 2016) in the equatorial Pacific, where a major decline of ~40% in surface chlorophyll is associated with a nearly total collapse in diatoms.

## Results

**Temperature and chlorophyll anomalies in the Pacific Ocean.** In this study, we focus our attention on four defined regions within the Pacific Ocean that have shown prominent high-positive SST and related high-negative surface Chl anomalies during the last decade (Fig. 1a). Within the northeast Pacific, we focus on three regions. The first two are the Gulf of Alaska (GOA) and a wide open ocean area stretching along the CCS, to which we refer as the "ARC" region, resembling the expression of a subsection of the eastern pole of the Pacific Decadal Oscillation (PDO)[24]. Located between these two is our third region of study, the North Pacific Transition Zone (NPTZ), a highly productive area where winter chlorophyll levels declined to record low levels in 2014[17]. Our fourth area of investigation is the ENSO 3.4 region in the equatorial Pacific, where extremely high-SST levels were recorded in 2016. It should be noted that our results are constrained to waters deeper than 200 m (Methods), thereby targeting primarily open ocean and not coastal areas.

Anomalies are obtained from the data-assimilating model at a monthly resolution. The model output of surface chlorophyll represents the mean concentration of phytoplankton Chl-based biomass in the model upper mixed layer. The model assimilates daily satellite ocean-color data, yielding improved biogeochemical output with respect to the original (unassimilated) model solution and correcting sampling errors in satellite retrievals[25] (Methods). Mean Chl anomalies throughout the North Pacific between 2013 and 2020 are mostly negative, with the exception of GOA (Fig. 1a). Averaged SST anomalies during this period are positive in all four regions assessed in this study (GOA, ARC, NPTZ, and ENSO 3.4) (Fig. 1b). SST (and Chl) anomalies are obtained with respect to the time span of the assimilated satellite record (2002–2020) and tend to be of lower amplitude than those computed in previous analyses of the North Pacific heatwave, which are based on longer SST time series dating back to the

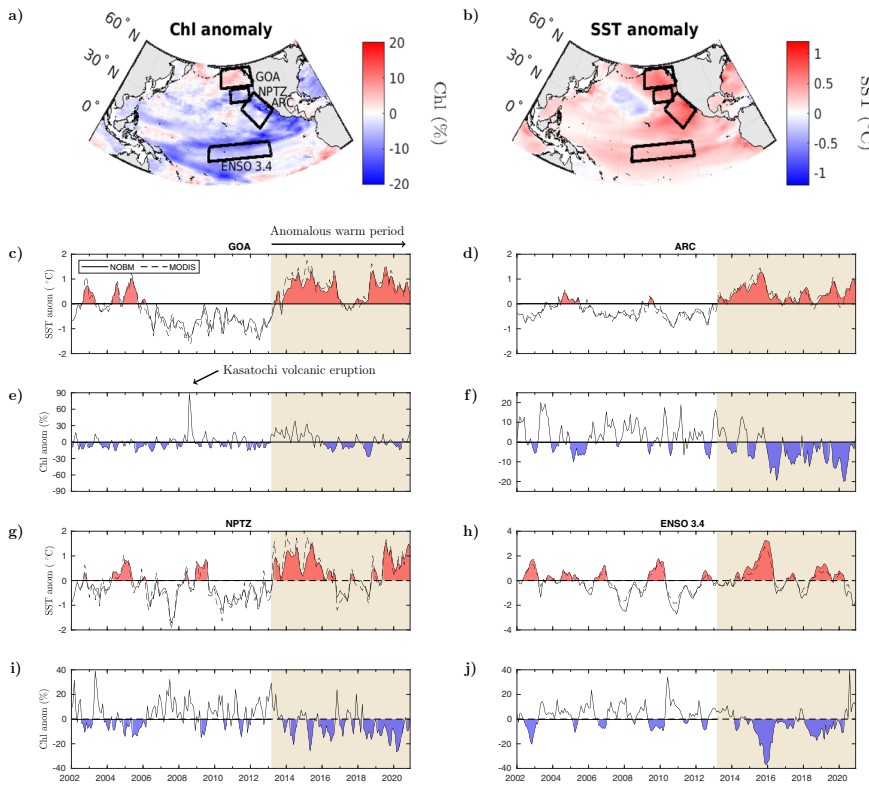

**Fig. 1 Chlorophyll (Chl) and sea surface temperature (SST) anomalies in the equatorial and northern Pacific. a, b** Mean modeled (**a**) Chl and (**b**) SST anomaly over the anomalously warm period between March of 2013 and December 2020. **c–j** Time series of monthly modeled Chl and SST anomalies between 2002 and 2020 for the four regions of the Pacific Ocean analyzed in this study: (**c**) GOA-SST, (**d**) ARC-SST, (**e**) GOA-Chl, (**f**) ARC-Chl, (**g**) NPTZ-SST, (**h**) ENSO 3.4-SST, (**i**) NPTZ-Chl, (**j**) ENSO 3.4-Chl. SST anomalies are shown in degree celsius (°C), and Chl anomalies are shown in percent change (%) relative to the deseasonalized climatological mean. Solid and dashed lines in SST plots are for NASA Ocean Biogeochemical Model (NOBM) output and MODIS Aqua retrievals, respectively. Area of beige background indicates the anomalously warm period initiated by the Blob heatwave in the northeast Pacific in March of 2013.

1880s[1,2] and include a reduced background component of anthropogenic warming in their baseline climatology. Over our assimilated time span, GOA presents two main periods of positive SST between 2002–2006 and 2013–2020 (Fig. 1c). Chl anomalies in GOA are mostly negative except for two main time intervals. The first and largest positive Chl anomaly is observed in August of 2008, where volcanic ash deposition from the Kasatochi eruption promoted one of the largest phytoplankton blooms recorded in the subarctic North Pacific[26]. This led to a nearly 90% increase in the data-assimilated output of Chl concentration (Fig. 1e). The second significant and more prolonged event of positive Chl anomalies in GOA is observed at the beginning of the anomalous warm period initiated by the Blob in 2013. Persistent positive Chl anomalies lasted for about three years, until 2016, and reached a high level of about 30% above the deseasonalized climatological mean. GOA-Chl anomalies declined to negative values after 2016, but the period of high-positive anomaly between 2013 and 2016 outweighs the average Chl anomaly output computed between 2013 and 2020 for this region (Fig. 1a).

Higher frequency and magnitude of concurrent positive SST and negative Chl anomalies is observed after 2013 in both the ARC and NPTZ regions (Fig. 1d, f, g, i). In the CCS region encompassed by ARC, Chl anomalies did not decline below −10% before 2013 (Fig. 1f). After 2013, Chl anomalies consistently declined below this threshold, reaching nearly −20%. Similarly, positive SST anomalies in ARC were far from reaching 1 °C before 2013, but surpassed this temperature threshold at the end of 2015 as the Blob expanded southwards within the North Pacific[1] (Fig. 1d). A comparable scenario is

observed in the NPTZ, where positive SST anomalies did not surpass 1 °C before 2013, but approximated a maximum SST anomaly of 2 °C between 2014–2016, and again in 2019–2020 (Fig. 1g). Negative Chl anomalies were constrained within −20% before 2013 in the NPTZ, but exceeded this negative threshold several times between 2013 and 2020, where lowest Chl levels were attained concurrently with highest SSTs (Fig. 1i) (note different Y-axis scales in panels of Fig. 1).

In the ENSO 3.4 region, SST and Chl anomalies are strongly anti-correlated, i.e., periods of warm, positive SST anomaly, mirror those of negative anomaly in surface Chl concentration (Fig. 1h, j). El Niño-Southern Oscillation (ENSO) events are known to dominate climate variability and its impact on ecological phytoplankton dynamics in this region[27]. The largest temperature increase between 2002 and 2020 signified a positive SST anomaly of ~4 °C during the 2016 El Niño and it is associated with a ~ 40 % decline in surface Chl, the largest decline registered in this region over the assimilated satellite record (Fig. 1j). Our results indicate that this wide decrease in Chl levels is linked with a transitory but major reorganization in phytoplankton community composition.

**Anomalies in phytoplankton functional types (PFTs).** The surface Chl concentration inferred from ocean optical properties retrieved via satellite remote sensing integrates information from the distinct phytoplankton communities present in the surface ocean. In our ocean biogeochemical model, total chlorophyll is obtained as the sum of the individual Chl-based biomass of six

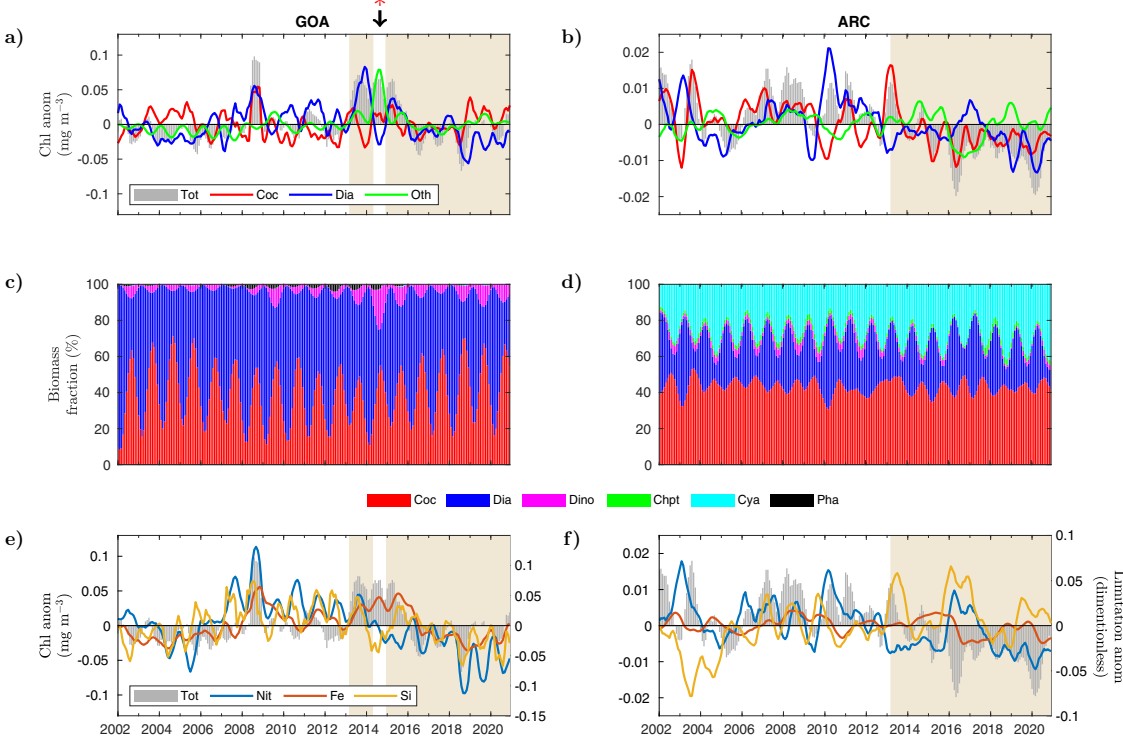

**Fig. 2 Monthly anomalies in phytoplankton functional types (PFTs) and nutrient limitation in GOA and ARC. a**, **b** (**a**) GOA and (**b**) ARC modeled anomalies (mg m$^{-3}$) in total surface chlorophyll (gray bars), coccolithophores (red line), diatoms (blue line), and dinoflagellates, cyanobacteria, chlorophytes, and phaeocystis combined together (summed) (green line). **c**, **d** Fraction of modeled total Chl biomass (%) corresponding to each PFT in (**c**) GOA and (**d**) ARC (coccolithophores-red, diatoms-blue, dinoflagellates-magenta, chlorophytes-green, cyanobacteria-cyan, and phaeocystis-black). **e**, **f** Anomalies in modeled surface chlorophyll (mg m$^{-3}$) (gray bars) and the growth-limiting term (unitless) of nitrate (blue), iron (red), and silicate (yellow) in (**e**) GOA and (**f**) ARC. Area of beige background indicates the anomalously warm period starting in March 2013. Red asterisk indicates the period between May and December of 2014 when noticeable changes in modeled PFT composition occurred in GOA. Monthly anomalies are smoothed using a 6-month rolling average.

PFTs, primarily distinct in their growth and sinking rates, as well as nutrient and light assimilation affinity (Tables S1, S2, and Fig. S1): diatoms, coccolithophores, chlorophytes, dinoflagellates, cyanobacteria, and phaeocystis. These six phytoplankton types encompass a wide range of carbon uptake and nutrient draw-down properties able to shape oceanic nutrient distributions and biogeochemical fluxes (Supplementary Note 1). This model partitioning allows us to decompose the time series of anomaly in total surface Chl into variations in the abundance of individual phytoplankton groups. The skill of the model in simulating the correct distribution of PFTs in the global ocean was evaluated against available in situ observations (Methods).

For this analysis, we applied a six-month rolling average to the monthly time series to discern the main patterns in the inter-annual variability of total Chl and that of the main PFTs (Figs. 2, 3). In GOA, the variability in total Chl is mainly driven by fluctuations in the concentration of diatoms and coccolithophores (Fig. 2a). These two groups tend to have opposite anomaly signs (i.e., diatoms have a positive anomaly when coccolithophores present a negative anomaly and vice versa) except during the Kasatochi volcanic eruption in the summer of 2008, when both groups display major positive anomalies. The integrated anomaly contribution of the remaining PFTs (green line, Fig. 2a) is of a lower magnitude throughout most of the analyzed period. The most notable exception to this pattern occurs about 18 months after the onset of the Blob in GOA, between May and December of 2014. During this period, the integrated anomaly of the remaining PFTs drives most of the temporal variability in total Chl. High-positive Chl anomalies after 2013 are

first sustained by a rapid increase in diatoms, and then by a rapid increase in the integrated concentration of the other groups (except coccolithophores). The initial increase in diatoms in 2013 is followed by a rapid decline in their surface concentration, which briefly alters the phytoplankton community composition in GOA.

The relative contribution of each PFT to the total phytoplankton Chl biomass remains rather stable in GOA between 2002 and 2020 (Fig. 2c). Diatoms (up to 80 % of biomass fraction) and coccolithophores (up to 70%) alternate seasonal dominance (i.e., occupy the largest fraction of total biomass) between spring and fall, respectively. The greatest ecosystem disruption occurs in summer–fall (May–December) of 2014, when dinoflagellates double their typical contribution to total Chl biomass from ~10% to 20%. This increase in the contribution of dinoflagellates to total Chl biomass is what drives the high-positive integrated anomaly in PFT's excluding diatoms and coccolithophores (green line, Fig. 2a). During this period the diatom fraction of total biomass decreased from normal levels ≥25% to ~10%.

Changes in the biomass fraction of PFTs in GOA are driven by variations in the supply and consumption of nutrients, primarily silicate and nitrate (Fig. 2e). We normalize nutrient anomalies by computing the growth-limiting term of the Monod-type function which depends on the actual nutrient concentration and the half-saturation concentration for each phytoplankton group (Methods). GOA anomalies in the nutrient limitation terms for silicate, nitrate and iron show similar inter-annual variability, coinciding in periods of positive and negative anomaly. The largest disagreement in the sign of anomaly occurs between 2014 and 2016, where nitrate and silicate concentrations decreased with

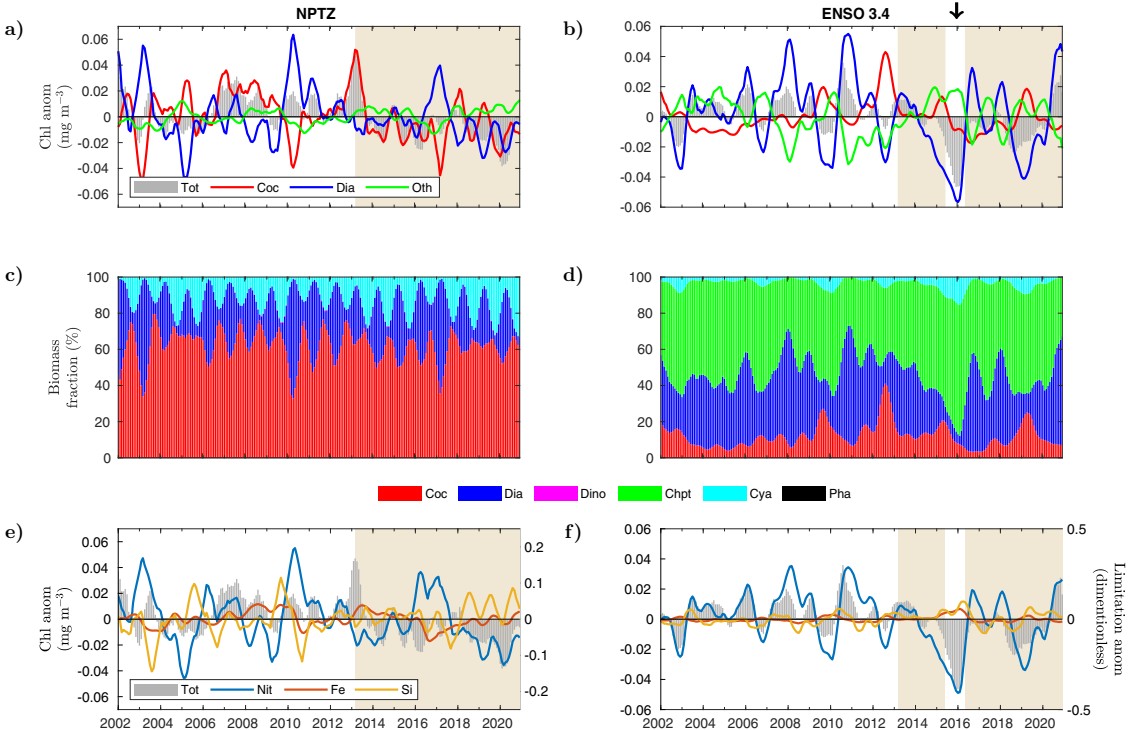

**Fig. 3 Monthly anomalies in phytoplankton functional types (PFTs) and nutrient limitation in NPTZ and ENSO 3.4. a, b** (**a**) NPTZ and (**b**) ENSO 3.4 modeled anomalies (mg m$^{-3}$) in total surface chlorophyll (gray bars), coccolithophores (red line), diatoms (blue line), and dinoflagellates, cyanobacteria, chlorophytes, and phaeocystis combined together (summed) (green line). **c, d** Fraction of modeled total Chl biomass (%) corresponding to each PFT in (**c**) NPTZ and (**d**) ENSO 3.4 (coccolithophores-red, diatoms-blue, dinoflagellates-magenta, chlorophytes-green, cyanobacteria-cyan, and phaeocystis-black). **e, f** Anomalies in modeled surface chlorophyll (mg m$^{-3}$) (gray bars) and the growth-limiting term (unitless) of nitrate (blue), iron (red), and silicate (yellow) in (**e**) NPTZ and (**f**) ENSO 3.4. Area of beige background indicates the anomalously warm period starting in March 2013. Red asterisk indicates the period between November of 2015 and March of 2016 when noticeable changes in modeled PFT composition occurred in ENSO 3.4. Monthly anomalies are smoothed using a 6-month rolling average.

respect to the deseasonalized climatological mean. During the expansion of the Blob (post 2013), negative anomalies are first observed in the silicate-limitation term during the later part of 2014, coinciding with the decline in diatoms. The anomaly in the nitrate-limitation term turns negative at the start of 2015, while iron concentrations increased, resulting in positive anomalies in the limitation term for this nutrient. This differing anomaly pattern is driven by a reduction in the supply of nitrate and silica, as well as a reduction in the consumption of iron by diatoms (see Discussion below).

In ARC, Chl anomalies averaged over a six-month rolling window switch from positive to negative values during 2013 and remain negative until 2020, indicating a sustained decline in Chl-based phytoplankton biomass until the end of the decade in this region (Fig. 2b). Diatoms and coccolithophores anomalies are mostly positive before 2013 and predominantly negative there-after. The integrated anomaly of all other groups is mainly positive between 2013 and 2020, except for the two year period between 2016 and 2018. However, no major alterations in the partitioning of PFTs are detected in this region over the assimilated satellite record (Fig. 2d). Coccolithophores (40–50% biomass fraction), diatoms (10–40%), and cyanobacteria (10–40 %) are the dominant groups in this region. The anomaly in the nitrate-limitation term is mostly positive before 2013 and negative thereafter, while the silicate-limitation term shows frequent positive anomaly after 2013 (Fig. 2f). The variability in the anomaly of the iron-limitation term has a narrower amplitude than for nitrate and silicate, with more frequent negative anomalous values after 2016 (Fig. 2f).

The NPTZ shows persistent negative Chl anomalies over the anomalous warm period (2013–2020) (Fig. 3a). In this region, negative Chl anomalies become more pronounced towards 2020, implying a further reduction in Chl-based phytoplankton biomass beyond the record low values observed in 2014[17]. In terms of the anomaly in phytoplankton groups, the NPTZ displays similar PFT anomaly dynamics as the previous two regions (GOA and ARC), with the largest fluctuations in the anomaly of diatoms and coccolithophores (Fig. 3a). As in GOA, these two phytoplankton groups tend to present opposite anomaly signs to each other. In the NPTZ, coccolithophores are the group that consistently occupies the largest total phytoplankton biomass fraction (40–80 %), followed by diatoms (5–60%) and cyanobacteria (15–30%) (Fig. 3c). No major disruptions in the dominance rank of these three groups is observed over the examined record. Similarly as in ARC, fluctuations in the anomaly of the normalized nitrate- and silicate-limitation terms are much wider than those in the iron-limitation term (Fig. 3e).

Out of the four examined regions, ENSO 3.4 registered the largest negative anomaly in Chl (~40%, Fig. 1j), which signified a reduction of nearly 0.06 mg Chl m$^{-3}$ (during the 2016 El Niño) with respect to the deseasonalized climatological mean for the 2002–2020 period (Fig. 3b). In this region, variations in total surface Chl biomass are largely driven by changes in the concentration of surface diatoms. The regions previously described above (GOA, ARC, and NPTZ) are located in mid-to high-latitude areas where strong seasonality in environmental conditions (e.g., nutrient and light availability) predominantly

dictates temporal changes in phytoplankton biomass fraction. The ENSO 3.4 region shows a much larger imprint of inter-annual variability in the partitioning of bulk Chl among PFTs (Fig. 3d). Here, chlorophytes are the dominant PFT (55–80% of biomass fraction), followed by diatoms (<5–60%), coccolitho-phores (5–40%), and cyanobacteria (<5–15%). As inferred from the decomposed Chl anomaly, diatoms are the group that experience the largest inter-annual variability in biomass partitioning. The ~40% decrease in total Chl biomass is associated with a nearly total collapse in diatoms from November 2015–March 2016. This event coincides with a warming anomaly of nearly 4 °C and a major reduction in surface nitrate concentration reflected in negative anomalies in the nitrate-limitation term (Fig. 3f). The variability in the anomaly of the iron- and silicate-limitation terms is much less pronounced with respect to that of nitrate. However, an increase in surface iron concentration driven by reduced diatom consumption permits the major shift in PFT structure observed during the 2016 El Niño event (see Discussion below).

## Discussion

We focus our discussion on the two regions that experienced the most significant changes in PFT composition over the assimilated (2002–2020) satellite record, GOA and ENSO 3.4. In GOA, the most important alteration took place between May and December of 2014, when the diatom fraction of total surface Chl biomass concentration drop to ~10% from levels typically ≥25% (Fig. 2). Nevertheless, bulk surface Chl levels remained above the desea-sonalized climatological mean, due to an increase in the surface concentration of dinoflagellates, which double their typical con-tribution to total Chl biomass from about 10% to 20%. At the core of these perturbations is a decline in the availability of surface silicate, an essential nutrient for diatom growth[28]. This phytoplankton group is characterized by constructing an outer silica cell wall (opal), which might serve as defense against grazers[29], and enhances their sinking[30,31]. The reduction in sur-face silicate is reflected in the negative temporal anomalies of the silicate-limitation term (Fig. 2e). The supply of nutrients to GOA is channeled by intermediate water from the subarctic marginal seas at the termination of the global circulation[32]. This "subarctic intermediate nutrient pool" (SINP) is advected from the western basin to the eastern subarctic Pacific and transported to the surface via wind-driven mixing and upwelling[32,33]. Inter-annual anomalies in Ekman vertical velocity estimated from the wind data forcing the circulation of the ocean biogeochemical model (Methods) agree well with anomalies in the modeled surface silicate concentration at GOA (Fig. 4a). Both surface silicate and Ekman velocities decline and show negative anomalies after 2013. Silicate anomalies remain negative between the start of the anomalous warm period and 2016. The anomaly in Ekman velocity is mostly negative during this period, except for a brief interval during 2014. Averaged over the duration of the Blob anomaly (March 2013–October 2015), Ekman velocities are weaker than normal over most of the GOA basin (Fig. 4c). The agreement between reduced wind-driven nutrient supply and modeled surface silicate levels is in accordance with one of the main trigger mechanisms of the Blob, in which weaker-than-average westerly winds induced anomalously weak Ekman transport of colder and nutrient-rich water[2]. The largest dis-agreement between the anomaly in surface silicate concentration and Ekman velocity is observed in August of 2008, where nutrient concentrations are corrected through the multivariate data assimilation to account for the enhanced productivity of organic matter triggered by the Kasatochi volcanic eruption detected in satellite observations.

In the ENSO 3.4 region, changes in community composition in 2015–2016 are driven by a stark reduction in the nitrate supply that fuels biological productivity in the equatorial Pacific (Figs. 3f, 4b). Nitrate supply in this region is tightly linked to variations in the upwelling and horizontal advection of nutrient-rich water associated to ENSO phases[34]. The El Niño-Southern Oscillation is the Earth's strongest source of inter-annual climate variability, highlighted by the exceptional 2015–2016 El Niño event[3,4]. The general mechanism by which El Niño events are set off is a weakening in trade winds that triggers a reduction in upwelling in the eastern Pacific, reinforcing warm SST anomalies over the equatorial Pacific through the Bjerknes feedback loop[35]. Changes in equatorial upwelling can be inferred from variations in SST, where warmer (positive) SST anomalies are associated to reduced upwelling and vice versa. Modeled SST anomalies in ENSO 3.4 are strongly anti-correlated to anomalies in modeled surface nitrate concentration (Fig. 4b, note inverted right Y-axis for SST). Over the assimilated record, the warmest SSTs and lowest nitrate concentrations are observed during the 2015–2016 El Niño, in which nitrate (SST) anomalies switch to negative (positive) values in February of 2014 and do not become positive (negative) until June of 2016. The lowest nitrate concentration is nearly 3 µ mol kg$^{-1}$ below the deseasonalized climatological mean, achieved in January of 2016. Mean SSTs anomalies between February of 2014 and June of 2016 are positive over most of the equatorial Pacific (up to 2°C warmer than normal, Fig. 4d) sug-gesting depressed upwelling around the ENSO 3.4 region. Con-trary to GOA, changes in PFT dominance in the equatorial Pacific are triggered by nitrate (instead of silica) limitation of diatom growth, resulting in the observed reduction of ~40% in surface Chl.

The initial decline in surface silicate in GOA during 2014 limits diatom growth and opens an ecological niche for dinoflagellates. In the biogeochemical model, the nitrate and iron requirements of dinoflagellates are the same as for diatoms (Table S1). How-ever, dinoflagellates do not require silicate to grow, a constraint exclusive to diatoms. Given their relatively high growth rate (90 % of diatoms' growth rate), dinoflagellates are placed in an ideal position to overtake most of the biomass fraction previously occupied by diatoms, especially as iron levels increased as a consequence of reduced diatom consumption. Iron deficiency is a limiting factor of phytoplankton growth in the northeast sub-arctic Pacific[36]. After 2016, the GOA surface concentrations of all nutrients represented in the model (nitrate, silicate, and iron) declined, as reflected in their limitation terms (Fig. 2e). This decline in nutrient concentration is likely driven by a reduction in wind-driven SINP transport represented by negative anomalies in Ekman velocity at the end of the decade (Fig. 4a), and is con-sistent with the resurgence a Blob-like SST anomaly in 2019[5]. The decline in the overall surface nutrient concentration is associated with a reduction in bulk surface Chl, but no clear disruption in the general composition of the phytoplankton community is observed, as all PFTs experience a similar loss in biomass.

Our model-based results coincide with several field observa-tions taken during the expansion of the Blob heatwave around GOA. Mean modeled silicate and diatom concentrations between May and December of 2014 are lower than normal over most of the area (Fig. 5a, c). Conversely, the averaged modeled iron concentration over the same time span is higher than the cli-matological mean (Fig. 5b), fueling the growth of dinoflagellates over most of the northern section of GOA (Fig. 5d). In situ samples taken from Continuous Plankton Recorders (CPRs) in the northern part of the basin (53°N–58°N, 136°W–146°W) indicate a decline in diatoms and increased abundance of dino-flagellates in the warm years of 2013–2014, although the overall abundance of diatom cells remains larger than that of

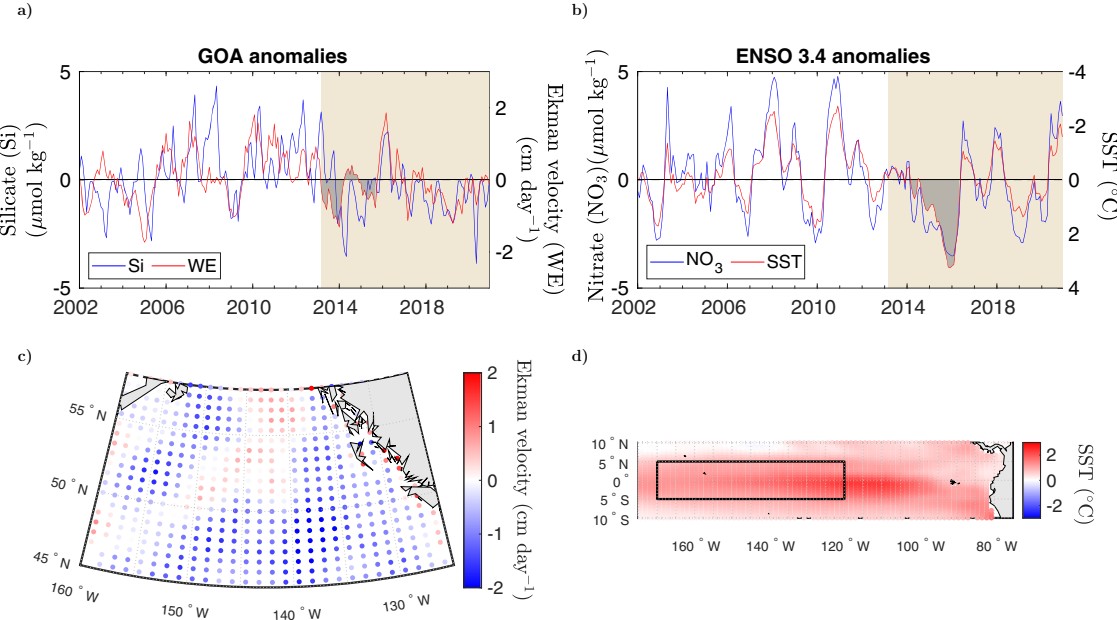

**Fig. 4 Wind-driven transport and nutrient anomalies. a** GOA monthly anomalies in modeled surface silicate concentration (μmol kg⁻¹) and Ekman vertical velocity (WE) (cm day⁻¹). Gray-shaded area in the silicate time series indicate the period of duration of the Blob between March of 2013–October 2015. **b** ENSO 3.4 monthly anomalies in modeled surface nitrate concentration (μmol kg⁻¹) and sea surface temperature (SST) (°C). Gray-shaded area in the nitrate time series indicate the period of duration of El Niño between February of 2014 and June of 2016. **c** Mean anomaly in Ekman vertical velocity over GOA between March of 2013–October 2015. **d** Mean anomaly in SST over ENSO 3.4 (black line rectangle area) between February of 2014 and June of 2016.

dinoflagellates[23]. Shipboard samples collected off Newport, Oregon (Newport Hydrographic Line, 44.6°N), showed increased dinoflagellate diversity and average abundance in samples associated to warm water masses from September 2014 through 2015[20]. While the Newport station is located near the GOA domain, these observations belong to waters no deeper than 60 m depth and need to be interpreted carefully as our model resolution does not include waters shallower than 200 m. Field samples along Line P in the northeast Pacific (extending from the southern tip of Vancouver Island to Ocean Station Papa (50°N 145°W) and covering the meridional range between 48°N–50°N) show contrasting results between inshore and offshore waters[21]. Haptophytes, a taxonomical algae group that includes coccolithophores, were observed to increase in Spring of 2014 and 2015 in the transition zone stations between inshore and offshore waters off the coast of North America (127°W–137°W). This is consistent with the increase in coccolithophores predicted by the model in the non-coastal eastern section of the basin (Fig. 5g). The most severe change reported along Line P occurred in the summer of 2015 at inshore stations, where both a strong decline in haptophytes and a accompanying increase in cyanobacteria were observed[21]. Model-based anomalies show a similar spatial pattern for these two groups during May–December 2014, where, on average, coccolithophores declined closer to the eastern coastal edge of the basin, while the cyanobacteria concentration increased in most of the eastern side of the basin and close to the coastline (Figs. 5e, S2e). The absolute increase in cyanobacteria Chl concentration (mg Chl m⁻³) is minor compared to that of dinoflagellates and coccolithophores, but their relative biomass rise is noticeable along the eastern section of GOA (Fig. S2e). Relative changes in modeled surface biomass of chlorophytes and phaeocystis are also discernible (Fig. S2f, h), but the magnitude of these variations in Chl concentration is negligible when compared to mean anomalies in the Chl fraction of the other PFTs, which are on the order of (±) 0.01 to 0.05 mg Chl m⁻³ (Fig. 5f, h). At

the iron-limited offshore stations of Line P, in situ Chl levels in 2014 were among the highest observed and comparable to the bloom levels induced by the natural volcanic fertilization of the Kasatochi eruption. Since there is no clear evidence of an increase in iron supply during the Blob, Peña et al.[21] hypothesized that elevated surface Chl concentrations could be due to reduced grazing and/or reduced light limitation as a result of enhanced surface stratification driven by the warm SST anomaly. Our results provide an alternate hypothesis, where the increase in total Chl during the summer–fall of 2014 is due to a shift in community composition in which higher than normal iron concentrations are achieved due to reduced diatom uptake (limited by diminished silicate supply), thereby fueling the growth of dinoflagellates, which partially overtake the Chl-biomass fraction normally occupied by diatoms.

The community composition in the ENSO 3.4 region differs considerably from that of GOA. During average climatological conditions, the equatorial tongue of cold water associated to high nutrient concentrations in the Pacific Ocean is mostly dominated by diatoms (Fig. S1). Chlorophytes dominate the outer edges of this tongue, while cyanobacteria occupy the oligotrophic subtropical gyres and the western section of the tropical Pacific. During the 2015–2016 El Niño perturbation, the mean modeled concentration of nitrate and diatoms decreased over most of the ENSO 3.4 region (Fig. 6a, c). Dinoflagellates also present negative relative biomass anomalies with respect to their climatological mean (Fig. S3d), but this reduction is negligible when compared to changes in the Chl-based biomass of the other groups (also on the order of (±) 0.01 to 0.05 mg Chl m⁻³) (Fig. 6d). Chlorophytes increased their abundance from east to west, forming a tongue-like shape similar to that of nutrient-rich waters advected from the eastern upwelling region (Fig. 6f). During the anomalous warm period induced by El Niño (November 2015–March 2016), this westward propagating tongue becomes a transition zone, in which nutrient availability is insufficient to allow diatoms to

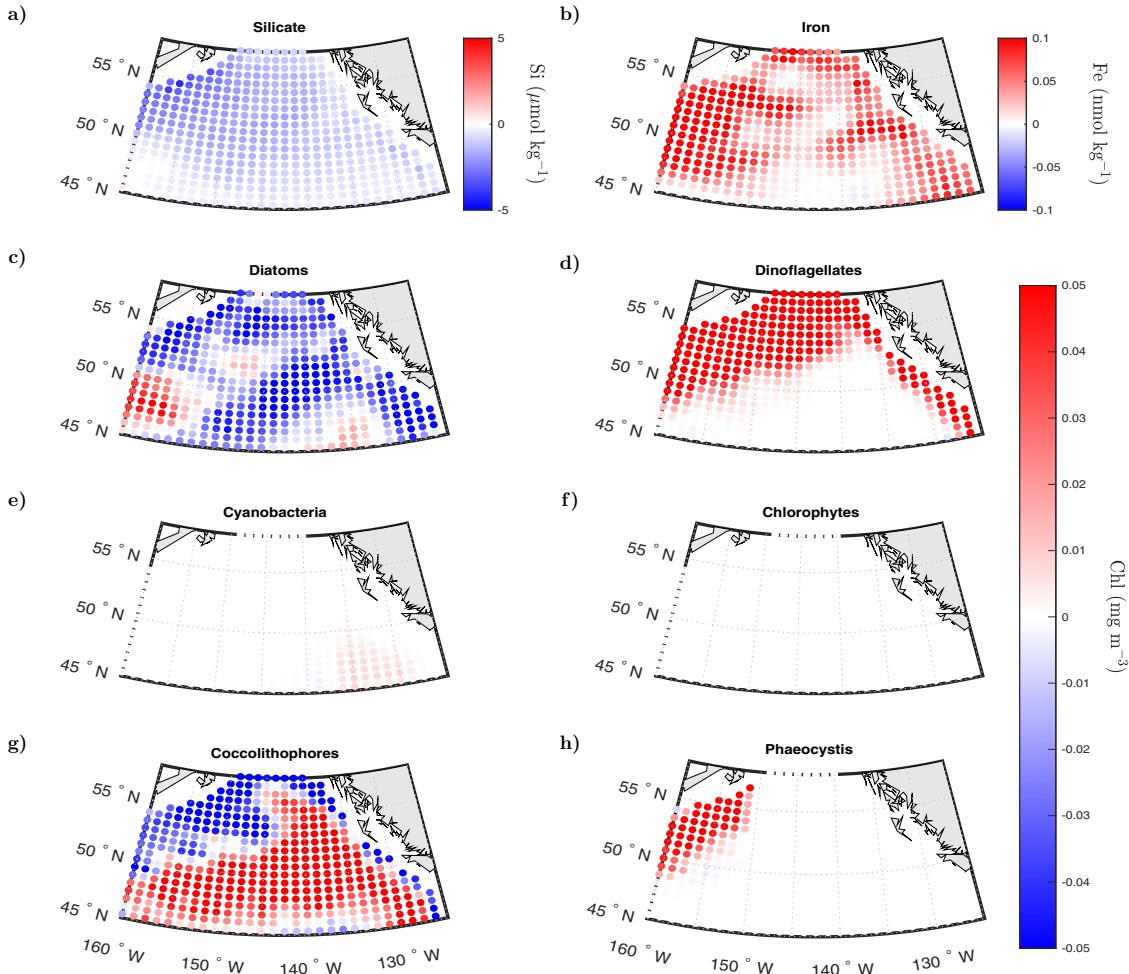

**Fig. 5 Spatial pattern in GOA nutrient and PFTs anomalies between May and December of 2014. a, b** Mean anomaly in modeled surface (**a**) silicate (μ mol kg⁻¹) and (**b**) iron (nmol kg⁻¹). **c–h** Mean anomaly in the modeled surface chlorophyll biomass (mg m⁻³) of (**c**) diatoms, (**d**) dinoflagellates, (**e**) cyanobacteria, (**f**) chlorophytes, (**g**) coccolithophores, and (**h**) phaeocystis.

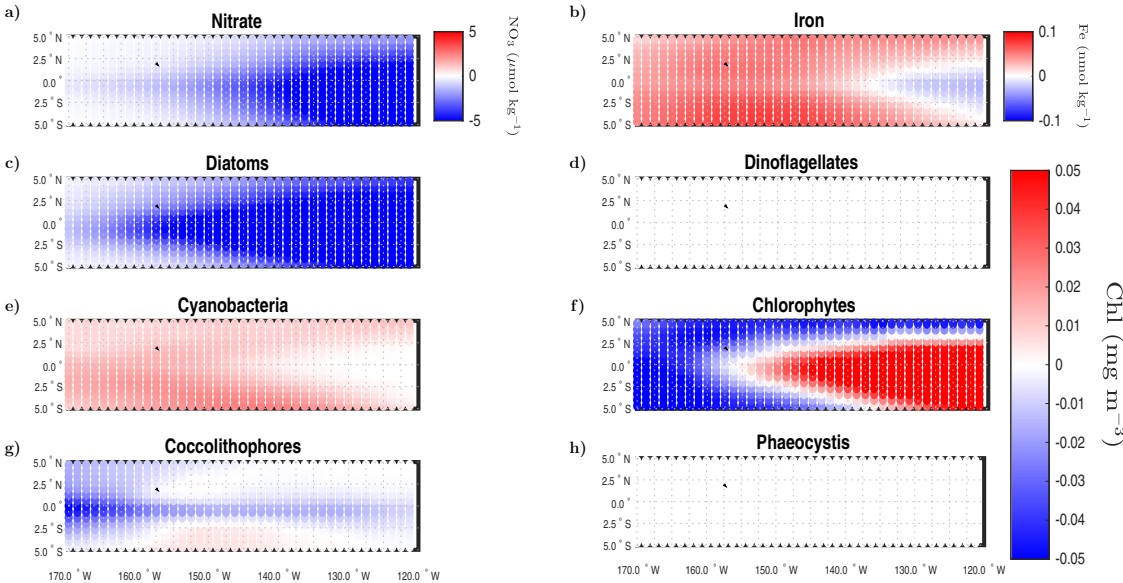

**Fig. 6 Spatial pattern in ENSO 3.4 nutrient and PFTs anomalies between November of 2015 and March of 2016. a, b** Mean anomaly in modeled surface (**a**) nitrate (μ mol kg⁻¹) and (**b**) iron (nmol kg⁻¹). **c–h** Mean anomaly in the modeled surface chlorophyll biomass (mg m⁻³) of (**c**) diatoms, (**d**) dinoflagellates, (**e**) cyanobacteria, (**f**) chlorophytes, (**g**) coccolithophores, and (**h**) phaeocystis.

predominate but still high enough for chlorophytes to out-compete other groups such as cyanobacteria. The overall diminishment in diatoms results in higher than normal iron levels over most of the ENSO 3.4 region, except along the westward extending tongue (Fig. 6b), where enhanced chlorophytes growth depletes surface iron concentrations with respect to their deseasonalized climatological mean. Mean positive anomalies in cyanobacteria biomass are observed outside this tongue (Fig. 6e), which benefit from enhanced iron availability and reduced competition from PFTs dependent on high macronutrient conditions. Anomalies in the surface concentration of coccolithophores and phaeocystis are noticeable in relative biomass terms (Fig. S3g, h), but negligible with respect to variations in the Chl concentration of the other groups (Fig. 6g, h).

The near total collapse in the surface concentration of diatoms predicted by our data-assimilating framework in early 2016 could have had major biogeochemical implications yet to be diagnosed by in situ surveys. Blooms of diatoms facilitate the vertical export and transfer of organic carbon and biochemical energy to higher trophic levels[37,38]. An immediate consequence attributed to the decline in diatoms is a reduction of about 40% in mean monthly surface Chl. To date, there are no available field observations of changes in phytoplankton community composition in the equatorial Pacific during the 2015–2016 El Niño. However, our model-based results simulate historical patterns associated with the biogeochemistry of this region. Consistent with previous observations[39,40], our results indicate that of all PFTs, diatoms are most responsive to inter-annual changes in nutrient conditions attributed to ENSO variability, even though this group is not the predominant contributor to total biomass. Fluctuations in the estimated contribution of diatoms to total Chl biomass varied from <20% during El Niño conditions in November of 1997 to >50% during La Niña (high nutrient) conditions in October 1998[41], which is similar to the modeled range over the assimilated satellite record (Fig. 3d). Despite frequently active upwelling periods that deliver high nitrate concentrations to the surface ocean, the modeled ENSO 3.4 region is dominated by chlorophytes, a group smaller in cell size than diatoms and more competitive in transition regions of low to high nutrient conditions[42]. This is consistent with observations that characterize the equatorial Pacific as a High Nutrient Low Chlorophyll (HNLC) region, where iron limitation[43–45] and silicate regulation[46] prevent full macronutrient utilization and corresponding higher productivity rates associated with diatom dominance.

A clear trend towards warmer SST and lower surface Chl levels is evident in the three examined regions of the North Pacific (GOA, ARC, and NPTZ) over the 2002–2020 assimilated satellite record (Fig. 7a, c, e). In ENSO 3.4, the most extreme levels of compound high-temperature and low-Chl were achieved in 2016 during the El Niño event, and returned to mean climatological conditions towards the end of that decade (Fig. 7g). These two areas, the northern and equatorial Pacific, are among the most prone to extreme high-temperature and low-Chl compound events in the global ocean, at least in the last 20 years[47]. Satellite observations indicate that bulk surface Chl concentrations are susceptible to conditions related to warm ocean anomalies, particularly in the northern and equatorial Pacific[17,47]. However, our results suggest that the community composition of PFTs in the North Pacific tends to be rather stable and resilient to environmental changes associated to marine heatwaves (except for the alterations observed during May–December of 2014). In the three regions of the North Pacific (GOA, ARC, and NPTZ), diatoms and coccolithophores predominate over other PFTs, even during high-SST and low-Chl anomalies (Fig. 7b, d, f). This highlights the strong control of the seasonal cycle in surface irradiance and

mixed-layer depth on surface nutrient supply and the median light level determining phytoplankton division rates[48]. Coccolithophores dominate during extreme SST conditions of the seasonal cycle, where their low light saturation for growth permits them to efficiently use reduced irradiance levels when vertical mixing associated to cold winter waters is high, while their low nutrient requirement to growth (i.e., half-saturation concentration) enables them to outcompete other groups in warm, late summer waters when surface nutrients are near depletion[49]. Conversely, diatoms dominate in the less extreme temperature conditions of spring and early summer, when relatively high nutrient availability and irradiance levels lead to the most optimal growth conditions of the annual seasonal cycle, driving the highest surface Chl biomass levels.

Environmental perturbations need to be of a greater magnitude than those imposed by the natural climate variability of the seasonal cycle, or create a unique imbalance in order to elicit a clear change in the phytoplankton community composition. For example, in May–December of 2014 the initial reduction in silicate surface concentration uniquely affected diatoms and opened a niche for dinoflagellates to grow in GOA. The dominance of dinoflagellates in certain areas (grid cells) of GOA is observed at SST anomalies ~1.5 °C and Chl anomalies > 0.4 mg Chl m$^{-3}$ (Fig. 7b). An increased dominance of cyanobacteria is observed towards high-SST and low-Chl conditions in certain grid locations within ARC (Fig. 7d) and NPTZ (Fig. 7f). This group have slow growth rates but high nutrient uptake efficiency, which allows them to thrive under oligotrophic nutrient conditions. While cyanobacteria did not clearly assume a presiding portion of the total Chl biomass fraction when averaged over the entire domain in neither of these two regions, they could become predominant under more severe warm anomalies linked to further reduced nutrient concentrations.

An abrupt change in PFT hierarchy is observed in ENSO 3.4 (Fig. 7h): diatoms dominate the regime of low-SST and high-Chl anomalies, while chlorophytes dominate the opposite extreme of compound high-SST and low-Chl anomalies. As extreme El Niño events become more frequent[50,51] and potentially more acute, the phytoplankton community composition could shift more permanently towards groups associated with less efficient productivity, transfer, and export of organic carbon. This shift could repercute on higher trophic levels, leading to less zooplankton abundance[23] and increased food stress on pelagic populations relevant to fisheries[52]. Persistent ecosystem responses to marine heatwaves are now starting to be identified in continental shelf waters[53]. In situ monitoring programs should complement satellite observations and modeling efforts to detect persistent ecological changes in areas of major economical and societal importance, such as the equatorial Pacific[54].

The validity of our results relies on the ability of the model to correctly simulate marine biogeochemical fields (e.g., surface Chl, relative PFT composition, and nutrient distributions) (Methods) and the extent to which these modeled fields are driven by the same main processes as in the real physical environment (i.e., correct mechanistic attribution). Our results indicate that large scale changes in phytoplankton community composition during the Pacific Ocean warm anomalies are mostly driven by "bottom-up" processes, where the availability of nutrients was altered by changes in their physical supply and differential consumption by the various PFTs. The NASA Ocean Biogeochemical Model (NOBM) has a thorough representation of the "bottom-up" processes that regulate phytoplankton growth (division) rates, but simplifies "top-down" control on biomass accumulation by assuming that grazing on phytoplankton occurs proportionally to the relative abundance of PFTs (Methods). In GOA, diatom biomass decreased due to reduced silicate availability associated to suppressed vertical

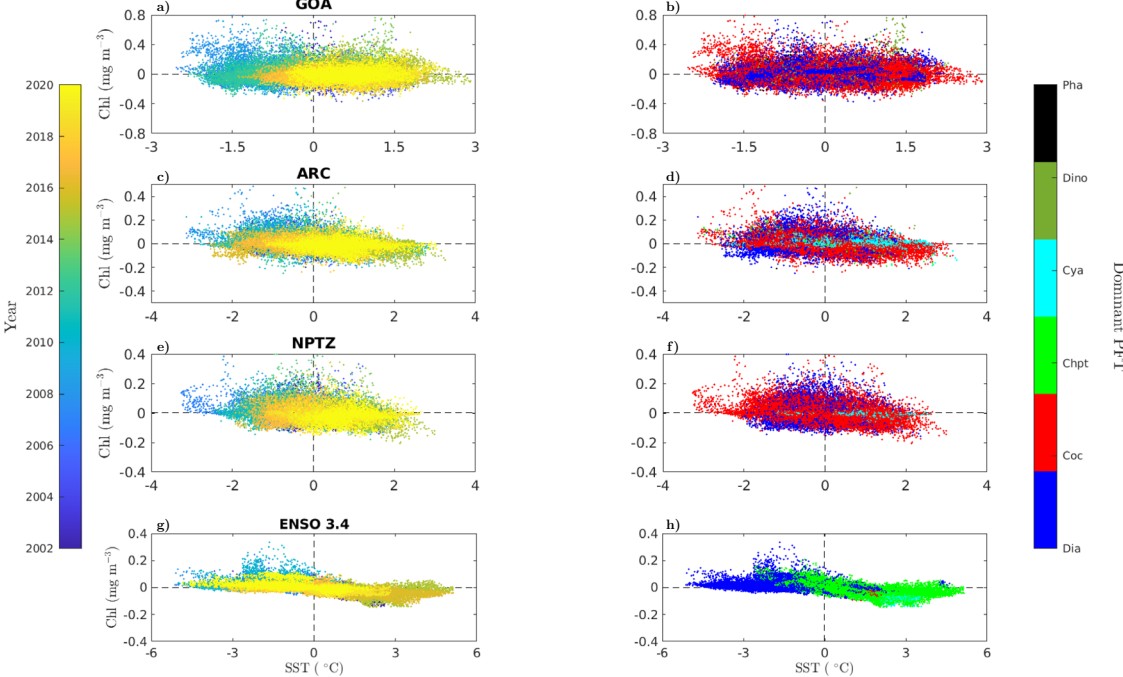

**Fig. 7 Relationship between compound chlorophyll (Chl) and sea surface temperature (SST) anomalies and phytoplankton group dominance.**
Scatterplots of modeled Chl (mg m$^{-3}$) and SST (°C) anomalies in (**a**, **b**) GOA, (**c**, **d**), ARC, (**e**, **f**), NPTZ, and (**g**, **h**) ENSO 3.4. Each data point corresponds to a monthly grid-cell output within the specified region. The color scale of panels in the left column indicates the corresponding year of each monthly data point between 2002 and 2020. The color scale of panels in the right column indicates the dominant phytoplankton functional type (PFT) (i.e., group that occupies the largest fraction of total Chl biomass) in a given monthly grid-cell output within the specified region (diatoms-blue, coccolithophores-red, chlorophytes-light-green, cyanobacteria-cyan, dinoflagellates-dark-green, phaeocystis-black).

nutrient supply and shallower mixed-layer depth during warm SST anomalies (Figs. 2, S4). Shoaling of the oceanic surface mixed layer can also enhance phytoplankton loss rates by increasing the predator-prey encounter rate and promoting grazing[55,56]. Under its current formulation, the NOBM is unlikely to detect or attribute changes in PFT composition due to such type of "top-down" regulation. However, our proposed mechanistic interpretation rooted primarily in alterations in nutrient availability due to reduced SINP tranport is consistent with the physical changes diagnosed during the North Pacific Blob[1,2,17] and driven by an observationally-based climatological reanalysis product[57]. In the ENSO 3.4 region, shallower mixed-layer depths are instead concomitant with cool SST anomalies and high nitrate concentrations in association with a reduced depth of the oceanic thermocline (Fig. S4). Here, the overall decrease in phytoplankton chlorophyll biomass observed in the satellite and model output during the strong 2016 El Niño, as well as the strong decline in diatoms predicted by the model, are consistent with enhanced nutrient limitation of phytoplankton growth and not increased grazing due to mixed-layer shoaling.

The depth-resolved light field driving the model is not stored among the suite of physical and biogeochemical variables derived from the NOBM. However, changes in light limitation after 2010 were examined by reconstructing the mixed-layer light field from the below surface irradiance inferred by the radiative transfer model (Methods). In GOA, light limitation is reduced (i.e., positive anomalies) (Fig. S5) due to shoaling of the mixed layer during the period of observed changes in PFTs composition (May–December 2014). In ENSO 3.4, anomalies in the light growth-limiting term are positive (i.e., reduced light limitation) during the 2016 El Niño (Fig. S5), as the strong decline in phytoplankton Chl biomas lead to a reduction in light attenuation, rising the overall light level in the mixed layer.

Our analysis is focused on the detection of shifts in phytoplankton community composition over relatively large areas of

the open ocean. These alterations are better diagnosed by applying a 6-month rolling average filter to the model monthly time series. The detection of these changes using a data-assimilation model framework should motivate the investigation of anomalies over a weekly/daily time scale and finer spatial resolution, which could reveal the preconditions for highly anomalous events of relevance for local ecosystems and fisheries, such as the development of toxic algal bloom. To this end, a more in depth mechanistic description of the marine food web typically represented in coupled ocean circulation-biogeochemical models might be needed.

## Methods

**General model description.** The model employed in this study is the NOBM. The NOBM provides a three-dimensional representation of ocean biogeochemistry processes coupled with a circulation and radiative transfer model[49]. The NOBM has a near-global domain spanning from 84°S to 72°N at a spatial resolution of $\frac{1}{4}°$ longitude and $\frac{2}{3}°$ latitude, and restricted to waters deeper than 200 m. Ocean circulation is rendered by the coupled Poseidon model[58], a reduced gravity model with 14 vertical layers in quasi-isopycnal coordinates. Physical forcing is obtained from the Modern-Era Retrospective analysis for Research and Applications version 2 (MERRA-2)[57,59] and includes wind speed and stress, SST, shortwave radiation, relative humidity, sea level pressure, sea ice fraction, cloud fraction and optical thickness. The light field is derived from the Ocean-Atmosphere Spectral Irradiance Model (OASIM)[60]. OASIM simulates the propagation of downward spectral irradiance in the atmosphere and ocean, and ultimately defines the level of photosynthetically available radiation (PAR) for phytoplankton growth. Access to NOBM software can be found at https://gmao.gsfc.nasa.gov/reanalysis/MERRA-NOBM/.

**Phytoplankton modeling.** Total chlorophyll (Chl$_{tot}$) is obtained as the sum of the Chl-based biomass of each of the six PFTs ($i$) represented in the model:

$$Chl_{tot} = \sum_{n=1}^{6} Chl_{(i)} \qquad (1)$$

The growth (division) rate ($\mu$) of each phytoplankton functional type is modeled as

a function of temperature ($T$), nutrients ($Nut$) and downwelling irradiance ($E$):

$$\mu_{(i)} = \mu_{max(i)} \; min\left[f(Nut_{(i)}), f(E_{(i)})\right] R \qquad (2)$$

where $\mu_{max}$ is the maximum phytoplankton growth rate and $R$ represents the temperature-dependent growth[61] normalized to 20 °C:

$$R = 1.066^{(T - 20°C)} \qquad (3)$$

The model accounts for limitation by nitrogen (N) ($NO_3 + NH_4^+$), iron (Fe), and silica (Si) (only for diatoms). Nutrient limitation is defined by the minimum of the Monod-type uptake function of these three nutrients:

$$f(Nut) = min\left[f(N, Fe, Si)\right] \qquad (4)$$

The Monod-type function is computed for each nutrient and PFT combination. This function is constrained by the half-saturation concentration ($k$), which is the nutrient concentration required by each phytoplankton group to achieve $\frac{1}{2}$ of the maximum (saturating) uptake rate for that given nutrient (Table S1). For example, the nitrate-limited growth fraction is defined as:

$$f(NO_3) = \frac{NO_3}{k_{NO_3(i)} + NO_3} \qquad (5)$$

where $k_{NO_3}$ is the nitrate half-saturation concentration. Light limitation is computed following a similar formulation:

$$f(E) = \frac{E}{k_{E(i)} + E} \qquad (6)$$

where $k_E$ is the irradiance at which $\frac{1}{2}$ of the light saturation parameter is achieved (Table S2).

Ecological loss terms of phytoplankton biomass include sinking ($w$) and herbivore grazing ($\gamma$). Sinking rates for each PFT are specified at 31 °C ($w_0$) (Table S1) and adjusted by temperature-dependent viscosity according to Stokes Law[62]:

$$w_{(i)} = w_{0(i)}(0.451 + 0.0178T) \qquad (7)$$

Grazing is modeled following an Ivlev formulation[63]:

$$\gamma = \gamma_m R_h \left[1 - \exp^{(-\Lambda \sum_{i=1}^{6} P_i)}\right] \qquad (8)$$

where $\gamma_m$ is the maximum grazing rate at 20 °C, $P$ is phytoplankton biomass and $R_h$ is the temperature dependence for grazing:

$$R_h = 0.06 \exp^{0.1T} + 0.7 \qquad (9)$$

Grazing represents the total loss of phytoplankton to herbivores, and is applied to the individual phytoplankton functional groups proportionately to their relative abundances. Therefore, top-down grazing pressure does not alter the relative composition of PFTs in the model. A complete NOBM description can be found in Gregg and Casey[49].

**Nutrient limitation and vertical velocity anomalies.** Nutrient limitation anomalies shown in Figs. 2e, f, 3e, f were computed based on the growth-limiting term of the Monod-type equation (e.g., Eq. 5). The $k$ value selected for each nutrient (nitrate, silicate, and iron) was that of diatoms (Table S1), given that this group tend to have the largest anomaly amplitude in the evaluated regions. The Ekman vertical velocity is derived from MERRA-2 surface wind speeds[57] and computed using the *ekman* function from the Climate Data Toolbox for MATLAB[64].

**Light limitation.** Due to computational constraints, the depth-resolved light field is not stored among the suite of physical and biogeochemical variables derived from the NOBM. However, we recomputed offline the light growth-limiting term experienced by phytoplankton in the mixed layer from the below ocean surface PAR estimated by OASIM and the mean mixed layer total chlorophyll concentration computed in the model. The PAR attenuation coefficient ($K_d$PAR) was empirically derived from the diffuse attenuation coefficient for downwelling irradiance at 490 nm ($K_d$490) following Eq. 9 of Morel et al.[65] ($K_d$PAR $= 0.0864 + 0.884 K_d490 - 0.00137 [K_d490]^{-1}$), while $K_d$490 is inferred from the model mean mixed-layer chlorophyll concentration following Eq. 8 in Morel et al.[65] ($K_d490 = 0.0166 + 0.0773[Chl]^{0.6715}$). The mean mixed-layer light level was retrieved using the Beer-lambert equation, and the light limiting term was computed using the Monod-type function (Eq. 6) using the half-saturation irradiance ($k_E$) defined for each light level in the model (Table S2). We recomputed the light growth-limiting term at GOA and ENSO 3.4 for the period 2010–2020, which covers the onset of the anomalous warm events starting in 2013 (Fig. S5).

**Data assimilation and model implementation.** The NOBM assimilated daily satellite ocean-color data of surface ocean chlorophyll, particulate inorganic carbon (PIC), and absorption by colored dissolved organic matter (aCDOM). The assimilation is primarily composed of data from the Moderate Resolution Imaging Spectroradiometer (MODIS) onboard of the Aqua satellite, part of NASA's Earth Observing System (EOS), and (to a minor degree) data from the Sea-viewing Wide

Field-of-view Sensor (SeaWiFS). The NOBM employs a multivariate sequential data-assimilation scheme in which assimilated model outputs (Chl, PIC, and aCDOM) are driven towards (satellite) observations through constant confrontation with data[66]. PFTs are not directly assimilated, but respond to changes in the physical environment that are affected by the assimilation of total satellite chlorophyll. The relative contribution of each PFT to total chlorophyll biomass is determined by the internal biogeochemical component of the model and preserved through the assimilation step. Nutrients are adjusted in accordance with the chlorophyll assimilation using nutrient-to-chlorophyll ratios embedded in the model. This adjustment can signify an important correction to the nutrient field in regions where the chlorophyll assimilation rectifies a persistent model bias[27] (e.g., the 2008 Kasatochi volcanic eruption). The model was initially spun up in free-run mode for 200 years using climatological forcing from MERRA[59]. The initial state for nitrate and silicate is taken from the World Ocean Atlas[67], dissolved iron is set from an ocean budget analysis[68], and ammonium is arbitrarily set to 0.5 μM. Phytoplankton concentrations are initialized to 0.5 mg Chl m$^{-3}$. Starting in 1980, the model is forced with monthly MERRA-2 data; the model begins assimilating satellite ocean-color data from SeaWiFS in January 1998. Output from this last step is used to initialize the assimilation of MODIS ocean-color retrievals in July 2002. Daily output from the SeaWiFS-assimilated run between January–June 2002 and from the MODIS-assimilated run between July 2002 and December 2020, is averaged monthly and analyzed in this study.

**Model validation.** Biogeochemical output from the NOBM has been extensively validated, including global nutrient and chlorophyll fields[49,69], as well as phytoplankton groups[27,70]. In this work, we conducted additional validation of the model surface Chl, PFTs, and nutrient output. First, we compare monthly anomalies in total Chl from the NOBM with satellite-based anomalies from MODIS and the Visible Infrared Imaging Radiometer Suite (VIIRS), and find an overall good agreement between model-based and satellite-based anomalies from both sensors for the 2002–2020 period (Fig. S6). Modeled surface Chl concentrations were validated against MODIS retrievals following the procedure in Gregg[66]: The percent error (PE) is calculated by computing the monthly median (*med*) chlorophyll concentration of the assimilation model (Chl$_{model}$) and MODIS (Chl$_{sat}$) over the entire domain where satellite retrievals and model output are co-located:

$$PE = \frac{med(Chl_{model}) - med(Chl_{sat})}{med(Chl_{sat})} \times 100 \qquad (10)$$

The median is used for error analysis as chlorophyll data tends to be log-normally distributed. The median is nearly independent of the data distribution and an easy-to-interpret metric of model bias. The annual error is computed for various oceanic regions as the mean of the monthly PE over the 12 months of the year. Results for the near-global model domain as well as the North Pacific, North Central Pacific, and Equatorial Pacific basins defined for model validation (Fig. S7) show that the model chlorophyll annual PE is constrained within ±15% between 2003 and 2020 (Fig. S8). Model and satellite chlorophyll fields are compared by matching the spatial resolution of the NOBM output to that of the satellite data and masking model output in accordance with gaps in the satellite retrievals.

The ability of the model to correctly estimate the contribution of PFTs to total chlorophyll is evaluated using a publicly available climatological database (https://gmao.gsfc.nasa.gov/reanalysis/MERRA-NOBM/data/phytogroups.xls). In situ observations on PFTs relative abundances are in general very scarce. We match up model mixed-layer phytoplankton relative abundances with the location and month of climatological in situ observations of the same type. Subsequently, we compute the mean relative contribution of each PFT to total Chl (PFT%) in each ocean basin where there is available climatological data and compare the contribution estimated by the model with that in the observations (PFT%$_{model}$ − PFT%$_{obs}$). In the Pacific Ocean, in situ observational data within the regions of interest in this study are mostly located in the North Pacific basin for GOA and the Equatorial Pacific for the ENSO 3.4 region (Fig. S7). Only a few observations are available in the North Central Pacific basin, but these are outside the defined ARC and NPTZ regions. Due to the scarce and sparse nature of the observations of PFT relative abundance in the Pacific Ocean, we rely on the global in situ dataset to assess the modeled relative abundance of PFTs by the NOBM (Table S3). Globally, the largest discrepancy in modeled relative abundance is found for cyanobacteria where the model underestimates their observed contribution to total Chl biomass by ~(−)15%. Within the pre-defined validation basins of interest in the Pacific Ocean, modeled PFTs abundances show the largest departure from observations for coccolithophores (+42%) in the North Pacific, and chlorophytes (+31%) in the Equatorial Pacific (Table S3). Differences between modeled and observed PFT relative abundance are reported for all model validation basins and all phytoplankton groups except dinoflagellates, for which there is insufficient available data (Table S3).

Modeled mean mixed-layer nutrient outputs for nitrate, silica, and iron were evaluated similarly as for chlorophyll, by calculating the PE of each model field with respect to observational data (Fig. S9). In situ nitrate and silica were obtained from the interpolated World Ocean Atlas[67]. Dissolved iron data were obtained from a compilation of 1951 observations created for the initial validation of the NOBM iron fields[69]. The PE was computed for the last year of the model simulation (2020) and reported for the near-global model domain as well as the

North Pacific, North Central Pacific, and Equatorial Pacific basins. Globally, the model tends to underestimate surface nitrate and iron by ~(−)24%, and silica by ~(−)46%. In the North Pacific, the model underestimates the observed mean nitrate and iron concentration by ~(−)3%, while the disagreement with in situ silica is of about −56%. In the North Central Pacific, the largest disagreement in nutrient fields is with respect to iron (~−66%), followed by silica (~−45%), and nitrate (~+15%). In the Equatorial Pacific, the model underestimates observed iron by (-) 55 %, while it overestimates observed silica by ~(+)40%, and nitrate by ~(+)15% (Fig. S9).

**Reporting summary**. Further information on research design is available in the Nature Portfolio Reporting Summary linked to this article.

## Data availability
Access to output from the NOBM can be gained via https://gmao.gsfc.nasa.gov/reanalysis/MERRA-NOBM/. The MERRA-2 reanalysis data used in this study have been provided by the Global Modeling and Assimilation Office (GMAO) at NASA Goddard Space Flight Center (GSFC), and can be accessed via https://gmao.gsfc.nasa.gov/reanalysis/MERRA-2/data_access/. Satellite ocean-color data can be obtained from the NASA GSFC ocean-color web site at https://oceancolor.gsfc.nasa.gov/. Modeled PFTs fields were evaluated using a publicly available climatological database available at https://gmao.gsfc.nasa.gov/reanalysis/MERRA-NOBM/data/phytogroups.xls.

## Code availability
NOBM code can be accessed via https://gmao.gsfc.nasa.gov/reanalysis/MERRA-NOBM/software/. The analyses presented here were conducted using the scientific programming software MATLAB (version 2022a).

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

## Acknowledgements
We thank the NASA Ocean Ecology Laboratory for providing the satellite ocean-color data and the NASA Center for Climate Simulation for computational support. We also thank the anonymous reviewers for the valuable feedback leading to the improvement of the paper. This paper was funded by the NASA Ocean Biology and Biogeochemistry Program (NNX16AR51G, Principal Investigator: C.S.R.) and the NASA PACE Science and Applications Team (80NSSC20M0208, Principal Investigator: C.S.R.).

## Author contributions
L.A.A. conducted the main scientific analyses and was the leading author of the paper. C.S.R. conducted the model simulations and contributed with the scientific analyses and discussion of the results.

## Competing interests
The authors declare no competing interests.
