## [Peer Review File · Communications Biology]

Reviewers' comments:

Reviewer #1 (Remarks to the Author):

This manuscript investigates the impact of marine heatwaves and extreme high-SST events in phytoplankton community composition. The authors focused on a set of different key regions of the North and Central Pacific Ocean and found that shifts in phytoplankton community composition (PFT composition and dominance) caused by high-SST events were evident and modulated by regional differences. The topic is very important and the conclusions of this kind of studies are likely to be foundational for future work. The manuscript is well written and clear.

I believe this work should be published and I look forward to seeing it published and cited but I have a few concerns that I should be addressed to put the findings of this study into context and make sure that readers are aware of the potential limitations of this study. These are mostly linked to the model the authors decided to use.

While any model has limitations and advantages and the authors' choice seems solid to me, at the moment the manuscript does not include enough reflections about what these limitations might be. 1) The authors state confidently that the model used, NOBM, has been widely validated. I think the authors should expand including more detail about what observations were used to expand such validation and, importantly, where. There is some information in the supplementary information, which I appreciate, but I think there should be more (for example, where is the data used for validation coming from? Maybe a map similar to Fig.3 of Gregg and Casey, 2007; but zoomed on the region of interest could be helpful) and that in the discussion some of the implications of the limitation of the model should be explored.

My familiarity with this model is limited, but I did dig a bit of the literature cited in this manuscript and I have found that, according to Gregg and Casey (2007), in Figure 3, the validation from in-situ sources is limited to the GOA and ENSO3.4 regions (using the labels of this study) but no in-situ observations appear to be used in the other subregions considered.

According to the text of the same manuscript, the North Pacific is a region where nitrates tend to be over-estimated (Fig.4 from Gregg and Casey, 2007) of at least a factor of 1.5, and diatom contribution to chlorophyll as well of a factor of at least 2.5 (Fig.6, Gregg and Casey, 2007). I'm wondering if this can have an impact in the diatoms-dinoflagellates contrast in the GOA. Would the same patterns be evident if there were less nitrates available? In the same paper, Gregg and Casey say: "Another region of disagreement between NOBM and data is the Equatorial Pacific, where the model indicated that coccolithophores dominated the phytoplankton at the distal (western) end of the upwelling region, beyond the area of dominance by diatoms."

I don't think these differences would dramatically impact the findings of this specific study as any bias would be present in both settings of extreme SST and "normal" temperatures, but there might be differences in what kind of difference in SST is needed to change the phytoplankton community composition, etc. and I think this limitation should be acknowledged.

The authors also cite the validation of the model carried out in Rousseaux and Gregg, 2012, where the model output is compared to the Hirata et al., 2011 method based on ocean colour images, which is quite re-assuring (within the limitations of satellite detection of PFT).

My overall suggestion is that in the manuscript we are assessing, the authors include a paragraph or two in the methods and the discussion where they add more detail about how the model was validated (in-situ and satellite, are more in-situ obs been added since the Gregg and Casey study?) and they discuss how the uncertainty associated with the validation could impact their results.

2) Another point that I think needs to be included in the discussion is that the model as it is includes a pretty solid description of bottom-up processes but it likely oversimplifies top-down processes and in

particular grazing. For example, all groups of phytoplankton are impacted by herbivory in the same way (proportionally to their abundance). The authors do point this out in the methods, but then in discussion I didn't find a good follow up about how this might impact the results of the study. I suspect that warmer SST could be associated to shallower mixed layers and this could affect the accessibility of phytoplankton to grazers (in terms of encounter rates, dilution and concentration) and this could potentially impact total chlorophyll but also PFT. For example, if diatoms are more resistant to grazing because of their silicon shells, they might deal better with increased grazing pressure associated to a shoaling of the mixed layer (although on the other hand, they might not get the silicates they need). I don't think the current model can address this question, but I do recommend that the authors discuss the potential impact of the assumptions related to grazing on their results. I would also consider adding a subfigure to Figure 4 showing changes in MLD as they could facilitate the discussion.

3) Something else that I wish was better described in the methods is the fact that the approach followed in this work is at very long temporal scales. I suspect that at least some of the events (fish mortality, toxic blooms) the authors describe in the intro to justify their work are likely to relate to higher frequency variability. While I understand that could be the scope for another publication, I do think the authors should discuss this point.

Minor points:

Line 513: Gregg et al., 2013 is a technical report from NASA. I wasn't sure if I accessed the right document (here? <https://core.ac.uk/download/pdf/10572994.pdf#page=54>) and if so I found that the description of the model wasn't as detailed as I hoped. Could you please include another reference (Maybe Gregg and Casey, 2007?) or the doi for a more comprehensive description? I also suggest pointing out here that there is more detail to be found in the supplementary materials as well.

Figure 2: I think the bright green to display the green algae might be a bit hard to see. In subfigure 2d) I had to squint really hard on my screen and on paper it was also quite hard. Maybe a darker shade of green would be more visible?

Line 171: A six-month rolling average seems to me to be an incredibly long-term average. How does this affect the results? Did the authors try shorter-timed rolling averages?

Reviewer #2 (Remarks to the Author):

Overview of the paper

This study investigates how heatwaves that have occurred over the past decade influence the structure of phytoplankton communities, expressed as Phytoplankton Functional Types (PFTs), in four regions of the Northeast and equatorial Pacific Ocean. To this end, the authors make use of a coupled ocean physics-radiative transfer-biogeochemical model with assimilation of satellite products (Chl, particulate inorganic carbon and light absorption by colored dissolved organic matter). Out of the four regions studied, two show significant changes in PFT composition and are therefore the focus of the analysis/discussion.

Comments

The document is nicely organized, reads well and addresses an interesting and timely topic. However, the analysis is quite qualitative and the discussion lacks strength so that the conclusions do not strike me as important. One might expect from such a model-based study to learn more about:

- (1) The mechanisms underlying the observed changes in PFT composition, from atmospheric forcing to modification of the stratification of the water column, nutrient supply and light conditions.
- (2) The impact of PFT changes with respect to the upper trophic levels and biogeochemical fluxes,

which are major implications cited in the introduction (« alter the transfer of energy throughout the food chain or the downward export of carbon » l. 75).^[1]

I am not sure what new and important information is provided by this study compared, for example, to the studies mentioned in the introduction (refs. 20-23) that are based on in situ (not modeled) data and that often used more quantitative approach to data analysis - e.g., Du & Peterson (2018) or Barth et al. (2020) who performed a clustering of their phytoplankton data in order to more clearly highlight how phytoplankton shifts are linked to environmental variability.

The model produces data that are inherently less reliable than in situ measurements. Yet, it may be used as a diagnostic tool (ex. through sensitive analysis) to study the chain of processes underlying changes in phytoplankton composition as well as their impacts on the biogeochemical fluxes (export in particular) and higher trophic levels. Forecasts could also be used to determine the resilience (or lack thereof) of ecosystems after the heat wave passes. Another interesting question could be the time lag between the onset of the positive temperature anomaly and when phytoplankton communities are actually affected by temperature-induced environmental changes. Hence, it seems to me that the model is not used to its full potential in the present study whose objectives would be better addressed with an in situ data set (as done in previous studies).

Specific comments (non-exhaustive)

Most of the analysis is based on visual examination of concurrent variations of anomalies in environmental drivers, limitation in nutrients, Chl and PFT composition (Figures 2 to 4), hence the mechanistic links between environmental drivers and changes in PFTs is not fully demonstrated. For example (l. 274-276 & Fig. 4a) the authors claim that « Inter-annual anomalies in Ekman vertical velocity... agree well with anomalies in the modeled surface silicate concentration at GOA ». This result is not obvious with the red (wind-induced transport) and blue (silicates) lines crossing each other (see, in comparison, the much clearer covariations between nitrates and SST in the ENSO region in Fig. 4b), and not pushed any further.

As another example, the authors point out a strong control of the observed changes in PFTs by light conditions in addition to nutrient limitation (l. 432-433 vs l. 435-437 and 441-442). But there is no analysis of deviations from the average light regime in the water column and how this might affect community composition. The coupled biogeochemical model used in this study is one of the few in the community, if not the only one, to have a light component (OASIM) and would therefore be a perfect tool to conduct to address such question.

The authors also argue that their results yield an alternate explanation to that of Peña et al. (2018) to explain the Chl anomaly associated with the Blob heatwave (l. 365-371). Whereas Peña et al. (2018) attributed the positive Chl anomaly to reduced grazing and/or reduced light limitation owing to increased stratification. Alternatively, the authors state that the changes in Chl result from changes in community structure associated with modification in the nutrient regime. I do not understand why the authors did not consider all the processes/parameters that are simulated by the model in order to decipher which from grazing, light, nutrient or community structure is the most important (and how they are interconnected with each other), a question that could not be addressed by, for example, Peña et al. based on field data. Again, I feel that the authors treated their model outputs as they would have treated "conventional" in situ data and thus did not get the most out of the modeling tool.

Another important point is the range of % anomalies in the PFTs compared with the range of % errors in the model data. Although the modeled PFT data could not be validated for each of the four study regions, it seems that the range of changes (anomalies in the relative contribution of the PFTs to Chl) are lower (ex. l. 187-196 for the GOA region; the trend is clearer for the ENSO 3.4 region) than the model error (Table S4). I think the authors should address this issue which is one of the keys of the strength of the analysis. More generally, the method used for validating the modeled PFT data is unclear. The Excel spreadsheet accessed from the URL given in p. 20 contains ~300 data points for

the surface or mixed layer. Some of the data appear to be cell counts, others pigment measurements. How is all this information transformed into the relative contribution of PFTs to Chl needed for model validation?

Reviewer #3 (Remarks to the Author):

Review of "Impact of Pacific Ocean heatwaves on phytoplankton community composition" by L.A. Arteaga and C.S. Rousseaux

Summary of manuscript:

This study uses an assimilating global scale ocean model with a biogeochemical component to investigate the impacts of recent Pacific Ocean heatwaves on the primary phytoplankton functional types that make up the phytoplankton community. Two events are the primary focus: the north Pacific "blob" event (with a focus on the Gulf of Alaska), and the 2015–2016 El Niño with a focus on the ENSO3.4 region of the Equatorial Pacific.

Overall Impression:

The paper is well written and the findings are interesting. Even though various studies are often categorized as "timely" this one actually fits that definition since there are at least a few observationally based studies of these two events to serve as a reality check.

I have one primary reservation with the study, and also include a few minor suggestions below.

My primary hang-up regards one of the two main findings: that dinoflagellates end up accounting for the increased chlorophyll in the GOA region during the "blob" event. I question whether this is simply an artifact of the assimilation and internal model dynamics, rather than reality? The question arises because while the model assimilates satellite chlorophyll estimates, the phytoplankton functional types (PFTs) have no real check and evolve according to internal nutrient and uptake rate specifications. That approach itself seems fine, yet from my reading there is no validation/constraint that exists for dinoflagellates, whereas the other PFTs have at least some basic validation. In a sense, the dinoflagellates represent a phytoplankton error term that will increase in the absence of diatoms in order to account for the elevated satellite chlorophyll that gets assimilated. Having stated that, the dinoflagellates in the model do tend to show up under warm (stratified) conditions with lower nutrients – this is nice to see, since that is generally speaking when they tend to accumulate in the real ocean. So at least the model is aimed at producing an expected result in that regard. Checking the relevant cited observational studies, I see very little evidence to suggest that dinoflagellates accounted for the enhanced chlorophyll signal observed during the "blob" event. Perhaps the best data set comes from the Line P observations (Pena et al. 2019), but no apparent dinoflagellate increase is documented there. The Newport observations are not appropriate for this global model, as discussed by the authors. The Continuous Plankton Recorder (CPR) observations suggests a modest uptick in dinoflagellate concentration, but the cell concentrations in Batten et al. (2022) are extremely low (4,000–8,000 cells/L), even during the "blob" years. In comparison, diatom concentrations were roughly 50,000 cells/L during that period, an order of magnitude larger than the concentrations of dinoflagellates. I suppose there is an open question of how well the CPR captures either of the quantities, but the observations suggest dinoflagellate concentrations that may be within the noise of diatom concentrations. So I guess the main question is: are there any other data sources out there that indicate an explosion of dinoflagellates during the "blob" that could be pointed to in order to lend credence to this finding? At present, the referred to data is not overly convincing. If no other such data exists, I have to question whether the authors need to re-pitch the GOA findings somehow – perhaps just as an error term? I'm honestly not sure what else to suggest but welcome the thoughts of the authors.

Minor comments/suggestions:

-Lines 32-34, awkward sentence, suggest rewording.

-Lines 77, 66, and 285 use the word notorious. Just my opinion but that's the kind of word you should probably just use once. It makes an impact when you read it the first time, but then you roll your eyes when you read it a second time. I suggest picking one of the sentences to use the word, and then choose a different descriptor for the other sentences.

-Fig. 1 – the SST time series show assimilative model and satellite SST, and how they agree/disagree. Can a similar approach be done with the Chlorophyll signal?

-Line 205, states "Negative anomalies are first observed... during the later part of 2014..." From Fig. 2e the silicate term is often negative prior to 2014, as is nitrate – e.g., from 2003–2006, but also nitrate turns negative prior to 2015. So perhaps this should be reworded?

Fig. 2. Currently the nutrient anomalies in Fig. 2e,f have the same colors as the PFTs above in Fig. 2a,b, and even c,d. Even though the figure subpanels are separated, this quickly gets confusing. I suggest that the line colors in Fig. 2e,f should be changed to some other independent colors just to help minimize confusion. Ditto for Fig. 3.

Fig. 3, and the discussion of the ENSO 3.4 region. Here Chlorophytes are abbreviated "Chl" and that is also used in the text for chlorophyll. These need to be differentiated better.

-Line 269, suggest deleting "towards higher depths".

-Fig. 4. Panels c, and d have axes labels that are way too small.

-Line 315, regarding iron and nitrate requirements... is this simply because of the division rate being different (the k values in Table S1 for diatoms and dinos are the same)?

-Fig. 7 right side panels, you might consider an alternative color for the dinoflagellates since magenta is difficult to see alongside red.

-Line 539, "bugget", do you mean "budget"?

-Regarding model validation. I don't find the chlorophyll "validation" all that useful, particularly a global annual average statistic. What does it even mean to validate an assimilative model? I suppose it just tries to quantify how well the model keeps what you gave it?

-Line 562, the link to climatological data needs updating

-Line 564, the PF groups in the Excel spreadsheet I found in relation to the bad link provided do not appear to have any months associated with them. So where is that information for evaluating the model? Perhaps I landed on the wrong spreadsheet?

Dear editors and reviewers,

Please find below our detailed responses to the comments made by the reviewers on the manuscript titled “Impact of Pacific Ocean heatwaves on phytoplankton community composition”, submitted for publication as an article to *Communications Biology*. We would like to take the opportunity to thank the reviewers for the effort they put into reviewing our manuscript, as well as for the valuable feedback leading to the improvement of the manuscript.

Reviewer (R) 1

General comments

1) R1: “This manuscript investigates the impact of marine heatwaves and extreme high-SST events in phytoplankton community composition. The authors focused on a set of different key regions of the North and Central Pacific Ocean and found that shifts in phytoplankton community composition (PFT composition and dominance) caused by high-SST events were evident and modulated by regional differences. The topic is very important and the conclusions of this kind of studies are likely to be foundational for future work. The manuscript is well written and clear.

I believe this work should be published and I look forward to seeing it published and cited but I have a few concerns that I should be addressed to put the findings of this study into context and make sure that readers are aware of the potential limitations of this study. These are mostly linked to the model the authors decided to use. While any model has limitations and advantages and the authors’ choice seems solid to me, at the moment the manuscript does not include enough reflections about what these limitations might be.”

Reply: Thank you for the overall positive comments on our manuscript. In the revised manuscript (rm), we have included a new section titled “Model caveats and considerations” (line 470), where we discuss the “bottom-up” orientation of the model in terms of mechanistic processes affecting changes in phytoplankton biomass, and the potential limitations of the NOBM to diagnose “top-down” controls related to grazing constraints on phytoplankton. We also highlight the orientation of our analysis towards open ocean and monthly time scales, and the necessity to augment the model resolution and mechanistic description of the planktonic food web to investigate more localized anomalous events (this section is also included below):

Model caveats and considerations

“The validity of our results relies on the ability of the model to correctly simulate marine biogeochemical fields (e.g., surface Chl, relative PFT composition, and nutrient distributions) (Methods) and the extent to which these modeled fields are driven by the same main processes as in the real physical environment (i.e.,

correct mechanistic attribution). Our results indicate that large scale changes in phytoplankton community composition during the Pacific Ocean warm anomalies are mostly driven by “bottom-up” processes, where the availability of nutrients was altered by changes in their physical supply and differential consumption by the various PFTs. The NOBM has a thorough representation of the “bottom-up” processes that regulate phytoplankton growth (division) rates, but simplifies “top-down” control on biomass accumulation by assuming that grazing on phytoplankton occurs proportionally to the relative abundance of PFTs (Methods). In GOA, diatom biomass decreased due to reduced silicate availability associated to suppressed vertical nutrient supply and shallower mixed layer depth during warm SST anomalies (Figure 2 and S4). Shoaling of the oceanic surface mixed layer can also enhance phytoplankton loss rates by increasing the predator-prey encounter rate and promoting grazing (Arteaga et al., 2020; Behrenfeld et al., 2013a). Under its current formulation, the NOBM is unlikely to detect or attribute changes in PFT composition due to such type of “top-down” regulation. However, our proposed mechanistic interpretation rooted primarily in alterations in nutrient availability due to reduced SINP transport is consistent with the physical changes diagnosed during the North Pacific Blob (Di Lorenzo and Mantua, 2016; Bond et al., 2015; Whitney, 2015) and driven by an observationally-based climatological reanalysis product (Gelaro et al., 2017). In the ENSO 3.4 region, shallower mixed layer depths are instead concomitant with cool SST anomalies and high nitrate concentrations in association with a reduced depth of the oceanic thermocline (Figure S4). Here, the overall decrease in phytoplankton chlorophyll biomass observed in the satellite and model output during the strong 2016 El Niño, as well as the strong decline in diatoms predicted by the model, are consistent with enhanced nutrient limitation of phytoplankton growth and not increased grazing due to mixed layer shoaling.

The depth-resolved light field driving the model is not stored among the suite of physical and biogeochemical variables derived from the NOBM. However, changes in light limitation after 2010 were examined by reconstructing the mixed layer light field from the below surface irradiance inferred by the radiative transfer model (Methods). In GOA, light limitation is reduced (i.e., positive anomalies) (Figure S5) due to shoaling of the mixed layer during the period of observed changes in PFTs composition (May–December 2014). In ENSO 3.4, anomalies in the light growth-limiting term are positive (i.e., reduced light limitation) during the 2016 El Niño (Figure S5), as the strong decline in phytoplankton Chl biomass lead to a reduction in light attenuation, rising the overall light level in the mixed layer.

Our analysis is focused on the detection of shifts in phytoplankton community composition over relatively large areas of the open ocean. These alterations are better diagnosed by applying a six-month rolling average filter to the model monthly time series. The detection of these changes using a data-assimilation model framework should motivate the investigation of anomalies over a weekly/daily time scale and finer spatial resolution, which could reveal the preconditions for highly anomalous events of relevance for local ecosystems and fisheries, such as the development of toxic algal bloom. To this end, a more in depth mechanistic description of the marine food web typically represented in coupled ocean circulation-biogeochemical models might be needed.”

2) R1: “The authors state confidently that the model used, NOBM, has been widely validated. I think the authors should expand including more detail about what observations were used to expand such validation

and, importantly, where. There is some information in the supplementary information, which I appreciate, but I think there should be more (for example, where is the data used for validation coming from? Maybe a map similar to Fig.3 of Gregg and Casey, 2007; but zoomed on the region of interest could be helpful) and that in the discussion some of the implications of the limitation of the model should be explored.

My familiarity with this model is limited, but I did dig a bit of the literature cited in this manuscript and I have found that, according to Gregg and Casey (2007), in Figure 3, the validation from in-situ sources is limited to the GOA and ENSO3.4 regions (using the labels of this study) but no in-situ observations appear to be used in the other subregions considered.”

Reply: We have revised the “Model validation” section in the main manuscript (line 608) and within the supplementary information to provide more clarity on how each of the different evaluated fields (nutrients, Chl, PFTs) were assessed. The newly included figure S7 (also Figure 1 in this reply document) shows the geographical location of in situ PFT observations (relative abundances) used for validation. This map has been updated with respect to that shown in Gregg and Casey (2007a) to include observations used for the validation of newly added PFTS, mostly in polar regions. As discussed in lines 637–650 of the rm, we rely on the global availability of observations to validate the model output on PFTs relative abundances: *“In the Pacific Ocean, in situ observational data within the regions of interest in this study are mostly located in the North Pacific basin for GOA and the Equatorial Pacific for the ENSO 3.4 region (Figure S7). Only a few observations are available in the North Central Pacific basin, but these are outside the defined ARC and NPTZ regions. Due to the scarce and sparse nature of the observations of PFT relative abundance in the Pacific Ocean, we rely on the global in situ dataset to assess the modeled relative abundance of PFTs by the NOBM (Table S3). Globally, the largest discrepancy in modeled relative abundance is found for cyanobacteria where the model underestimates their observed contribution to total Chl biomass by $\sim (-)$ 15%. Within the pre-defined validation basins of interest in the Pacific Ocean, modeled PFTs abundances show the largest departure from observations for coccolithophores (+ 42 %) in the North Pacific, and chlorophytes (+ 31 %) in the Equatorial Pacific (Table S3). Differences between modeled and observed PFT relative abundance are reported for all model validation basins and all phytoplankton groups except dinoflagellates, for which there is insufficient available data (Table S3).”*

3) R1: “According to the text of the same manuscript, the North Pacific is a region where nitrates tend to be overestimated (Fig.4 from Gregg and Casey, 2007) of at least a factor of 1.5, and diatom contribution to chlorophyll as well of a factor of at least 2.5 (Fig.6, Gregg and Casey, 2007). I’m wondering if this can have an impact in the diatoms-dinoflagellates contrast in the GOA. Would the same patterns be evident if there were less nitrates available? In the same paper, Gregg and Casey say: “Another region of disagreement between NOBM and data is the Equatorial Pacific, where the model indicated that coccolithophores dominated the phytoplankton at the distal (western) end of the upwelling region, beyond the area of dominance by diatoms.”

I don’t think these differences would dramatically impact the findings of this specific study as any bias would be present in both settings of extreme SST and “normal” temperatures, but there might be differences in what kind of difference in SST is needed to change the phytoplankton community composition, etc. and I

Figure 1: Distribution of in situ observations of PFTs relative abundances (i.e., percentage of Chl-based biomass relative to total chlorophyll) used for validation of the NOBM. Color scale indicates the total number of functional types observed at each location (out of the six PFTs represented in the model) (Figure S7 in rm).

think this limitation should be acknowledged.”

Reply: We have extended our validation analysis to include the model nutrient fields. The newly added Figure S9 shows that the current model configuration underestimates nitrate by about 3 % in the North Pacific, while the diatom contribution to total chlorophyll is overestimated by about 30 % in this region (Table S3). Both results represent an important improvement with respect the model version assessed in Gregg and Casey (2007a). Climatologically, phytoplankton biomass in the western end of the Equatorial Pacific is now dominated by chlorophytes and cyanobacteria, in line with low macronutrient distributions in this oceanic region (Figure S1). We do not think that any of these model–data discrepancies represent a consistent bias in the model, which is driven by meteorological reanalysis inputs (MERRA-2, Gelaro et al., 2017). Also, the overall climatological distribution of PFTs within the North and Equatorial Pacific Ocean is in line with the surface distribution in macro and micronutrients represented in the model (Figure S1).

4) R1: “The authors also cite the validation of the model carried out in Rousseaux and Gregg, 2012, where the model output is compared to the Hirata et al., 2011 method based on ocean colour images, which is quite re-assuring (within the limitations of satellite detection of PFT).

My overall suggestion is that in the manuscript we are assessing, the authors include a paragraph or two in the methods and the discussion where they add more detail about how the model was validated (in-situ and satellite, are more in-situ obs been added since the Gregg and Casey study?) and they discuss how the uncertainty associated with the validation could impact their results.”

Reply: Thank you for this suggestion. As indicated in reply 2, we have revised and extended the model validation section within the main manuscript as well as in the supplementary material. In the supplementary material, we have included subsections expanding on the methodology and results for the validation of surface chlorophyll, PFTs, and nutrient fields. We have also revised the presentation of these results, now including Figure S7, which shows the geographical distribution of in situ PFT data, Figure S9, which presents the model vs. data comparison of the nutrient fields, and Figure S8, which substitutes a previous table and shows in a clearer manner the range of uncertainty in modeled surface Chl with respect to MODIS.

5) R1: “Another point that I think needs to be included in the discussion is that the model as it is includes a pretty solid description of bottom-up processes but it likely oversimplifies top-down processes and in particular grazing. For example, all groups of phytoplankton are impacted by herbivory in the same way (proportionally to their abundance). The authors do point this out in the methods, but then in discussion I didn’t find a good follow up about how this might impact the results of the study. I suspect that warmer SST could be associated to shallower mixed layers and this could affect the accessibility of phytoplankton to grazers (in terms of encounter rates, dilution and concentration) and this could potentially impact total chlorophyll but also PFT. For example, if diatoms are more resistant to grazing because of their silicon shells, they might deal better with increased grazing pressure associated to a shoaling of the mixed layer (although on the other hand, they might not get the silicates they need). I don’t think the current model can address this question, but I do recommend that the authors discuss the potential impact of the assumptions related to grazing on their results. I would also consider adding a subfigure to Figure 4 showing changes in MLD as they could facilitate the discussion.”

Reply: We agree with the reviewer in that the current version of the NOBM is more complete in its description of bottom-up planktonic processes than top-down grazing controls. As indicated in reply 1, we have included a new section discussing model caveats and limitations (line 470e of the rm), which focuses primarily on mechanistic process attribution. In the revised manuscript, we have also added Figure S4 to assess the confounding effects of mixed layer shoaling on phytoplankton biomass. As discussed in the section “Model caveats and consideration”, we attribute the total phytoplankton biomass reduction in GOA to lower nutrient availability coincident with reduced vertical nutrient supply and shallower mixed layer depths, but we cannot rule out enhanced grazing from increased predator-prey encounters under these conditions. However, this mechanism would not apply in ENSO 3.4, where the strong decline in total phytoplankton biomass and diatoms observed in the satellite and model output during the strong 2016 El Niño occurs during relatively deeper mixed layer associated to reduced equatorial upwelling.

6) R1: “Something else that I wish was better described in the methods is the fact that the approach followed in this work is at very long temporal scales. I suspect that at least some of the events (fish mortality, toxic blooms) the authors describe in the intro to justify their work are likely to relate to higher frequency variability. While I understand that could be the scope for another publication, I do think the authors should discuss this point.”

Reply: Thank you for raising this point. As indicated in reply 1, we have included a brief discussion on the time scale of our analysis at the end of the “Model caveats and considerations” section (see line 470 of the rm). We hope that our detection of shifts in phytoplankton biomass and PFTs composition, which are in general agreement with observations, can serve as a foundation and motivation to investigate anomalous events through marine ecological models coupled with data-assimilation capabilities.

Minor points

7) R1: “Line 513: Gregg et al., 2013 is a technical report from NASA. I wasn’t sure if I accessed the right document (here? <https://core.ac.uk/download/pdf/10572994.pdf#page=54>) and if so I found that the description of the model wasn’t as detailed as I hoped. Could you please include another reference (Maybe Gregg and Casey, 2007?) or the doi for a more comprehensive description? I also suggest pointing out here that there is more detail to be found in the supplementary materials as well.”

Reply: The model description in the technical report of Gregg et al. (2013) (correct url: <https://gmao.gsfc.nasa.gov/pubs/docs/Gregg597.pdf>) is based on the more in depth description presented in Gregg and Casey (2007a). We refer to the latter in the rm (line 559).

8) R1: “Figure 2: I think the bright green to display the green algae might be a bit hard to see. In subfigure 2d) I had to squint really hard on my screen and on paper it was also quite hard. Maybe a darker shade of green would be more visible?”

Reply: Thank you for highlighting this. We have improved the visibility of all lines by increasing their size (a darker green shade was still difficult to see).

9) R1: “Line 171: A six-month rolling average seems to me to be an incredibly long-term average. How does this affect the results? Did the authors try shorter-timed rolling averages?”

Reply: Yes, we started our analysis without averaging the monthly-based model data, and also tried shorter rolling means. The six-month rolling average was able to minimize the short term variability of the time series while still accentuating the main changes in PFTs biomass during the anomalous warm period.

Reviewer (R) 2

1) R2: “Overview of the paper

This study investigates how heatwaves that have occurred over the past decade influence the structure of phytoplankton communities, expressed as Phytoplankton Functional Types (PFTs), in four regions of the Northeast and equatorial Pacific Ocean. To this end, the authors make use of a coupled ocean physics-radiative transfer- biogeochemical model with assimilation of satellite products (Chl, particulate inorganic carbon and light absorption by colored dissolved organic mater). Out of the four regions studied, two show significant changes in PFT composition and are therefore the focus of the analysis/discussion.”

“Comments

The document is nicely organized, reads well and adresses an interesting and timely topic. However, the analysis is quite qualitative and the discussion lacks strength so that the conclusions do not strike me as important. One might expect from such a model-based study to learn more about: (1) The mechanisms underlying the observed changes in PFT composition, from atmospheric forcing to modification of the stratification of the water column, nutrient supply and light conditions. (2) The impact of PFT changes with respect to the upper trophic levels and biogeochemical fluxes, which are major implications cited in the introduction (- alter the transfer of energy throughout the food chain or the downward export of carbon - l. 75). I am not sure what new and important information is provided by this study compared, for example, to the studies mentioned in the introduction (refs. 20-23) that are based on in situ (not modeled) data and that often used more quantitative approach to data analysis - e.g., Du & Peterson (2018) or Barth et al. (2020) who performed a clustering of their phytoplankton data in order to more clearly highlight how phytoplankton shifts are linked to environmental variability.”

Reply: Many thanks for reviewing our manuscript. The goal of our study was to investigate whether large scale changes in PFTs are diagnosed by a biogeochemical model constrained by satellite ocean color and forced by meteorological reanalysis products. The studies of Du and Peterson (2018) and Barth et al. (2020) provide important in situ evidence of alterations in coastal regimes, while our results highlight changes in open ocean regions, where water depth exceeds 200 m. We identify and describe changes in silica and nitrate availability as the main underlying drivers of alterations in PFTs assemblage in GOA and the ENSO 3.4 region. Throughout our discussion, we link changes in nutrient availability with alterations in physical circulation, namely advection of “subarctic intermediate nutrient pool” (SINP) in GOA, and ENSO dynamics in the Equatorial Pacific. The atmospheric drivers of these physical changes have already been described in previous studies (Di Lorenzo and Mantua, 2016; Bond et al., 2015; Blunden and Arndt, 2016; Santoso et al., 2017) and are beyond the scope of our work. We quantify the decline in total chlorophyll in absolute (mg Chl m^{-3}) and relative (%) units, and relative (%) alterations in PFTs, which has not been documented before. Our conclusions are based on model output that has been validated with available in situ data within and outside of the regions of focus of in our study. In our revised manuscript (rm) we have expanded the description of the validation tasks (Methods section) and improved the presentation of the validation results (Methods section and Supplementary Information). Changes in light availability were not initially investigated as depth-resolved light fields are not stored within the biogeochemical suite of NOBM variables. However, we recomputed the mixed layer light level from OASIM and found that changes in light limitation

during the heatwave periods are reduced compared with changes in nutrient limitation (see reply 4 below). The impact of changes in PFT on the cycling of nutrients, carbon export, and upper trophic levels are part of ongoing and future work, but we believe that it was necessary to first establish whether anomalies in oceanic PFTs occurred, and to describe their magnitude and main features. This is the primary goal of our submitted manuscript.

2) R2: “The model produces data that are inherently less reliable than in situ measurements. Yet, it may be used as a diagnostic tool (ex. through sensitive analysis) to study the chain of processes underlying changes in phytoplankton composition as well as their impacts on the biogeochemical fluxes (export in particular) and higher trophic levels. Forecasts could also be used to determine the resilience (or lack thereof) of ecosystems after the heat wave passes. Another interesting question could be the time lag between the onset of the positive temperature anomaly and when phytoplankton communities are actually affected by temperature-induced environmental changes. Hence, it seems to me that the model is not used to its full potential in the present study whose objectives would be better addressed with an in situ data set (as done in previous studies).”

Reply: We agree that in situ data is inherently less uncertain than model-based data. However, the value of model-based reanalysis products is that it allows the reconstruction of biogeochemical variables over regions and periods that were not originally sampled in situ. For our study, the NOBM was not run in forecast mode, but forced with climatological reanalysis data (MERRA-2, Gelaro et al., 2017). We agree that all the examples mentioned above represent interesting applications of mechanistic ocean biogeochemical models. However, exhaustive in situ sampling of changes in open ocean conditions was not conducted simultaneously with the warm anomalies observed during the last decade over the Pacific Ocean. A valuable alternative to assessing such changes is via the reconstruction of the biogeochemical fields using meteorological reanalysis and the available satellite record. The purpose of our study is to leverage the available NASA’s ocean color record through its assimilation in the NOBM in order to assess the impact of such warm anomalies on PFTs as a first step in exploring the biogeochemical ramifications of the marine heatwaves.

3) R2: “Specific comments (non-exhaustive)”

“Most of the analysis is based on visual examination of concurrent variations of anomalies in environmental drivers, limitation in nutrients, Chl and PFT composition (Figures 2 to 4), hence the mechanistic links between environmental drivers and changes in PFTs is not fully demonstrated. For example (l. 274-276 & Fig. 4a) the authors claim that - Inter-annual anomalies in Ekman vertical velocity... agree well with anomalies in the modeled surface silicate concentration at GOA -. This result is not obvious with the red (wind-induced transport) and blue (silicates) lines crossing each other (see, in comparison, the much clearer covariations between nitrates and SST in the ENSO region in Fig. 4b), and not pushed any further.”

Reply: In the NOBM, as well as most ocean Nutrient-Phytoplankton-Zooplankton-Detritus (NPZD) models, phytoplankton growth (division) rates (μ) respond immediately to environmental nutrient and light conditions following Eq 2 of the methods section ($\mu_{(i)} = \mu_{max(i)} \min[f(Nut_{(i)}), f(E_{(i)})]$ R). Hence, the

visual determination of the minimum growth-limiting term presented in Figure 2e-f and 3e-f does demonstrate the mechanistic link between environmental drivers and changes in PFTs, at least from a bottom-up perspective (i.e., it identifies the resource limiting phytoplankton division rates). As mentioned above, a missing component in these plots is light limitation, which we show below (reply 4) has a reduced impact on phytoplankton growth compared to changes in nutrient limitation in GOA and ENSO 3.4. A potential weakness of our analysis is the more simplistic model representation of phytoplankton “top-down” controls with respect to that of phytoplankton growth. In the rm, we have included a new section titled “Model caveats and considerations” (line 470), where we discuss the “bottom-up” orientation of the model in terms of mechanistic processes affecting changes in phytoplankton biomass, and the potential limitations of the NOBM to diagnose “top-down” controls related to grazing constraints on phytoplankton.

We verified the relationship between anomalies in Ekman vertical velocity (WE) and modeled surface silicate at GOA by computing the coefficient of determination (R^2) between monthly anomalies of both variables over the 2002 – 2020 period covered by the NOBM run and presented in Figure 4 of the rm (Figure 2 of this document). The obtained R^2 of 0.43 indicates that WE explains nearly half of the variability in surface Si, suggesting a strong mechanistic link between wind anomalies and nutrient delivery in GOA, similar to that proposed for the NPTZ (Whitney, 2015).

Figure 2: Top panel: Time series of GOA monthly anomalies in Ekman vertical velocity (WE) (cm day^{-1}) computed from MERRA-2 wind data (red line, left Y-axis) and modeled surface silicate concentration ($\mu\text{mol kg}^{-1}$) from the NOBM (blue line, right Y-axis). WE anomalies are computed for the entire MERRA-2 wind reanalysis record. Bottom panel: Scatter plot of monthly anomalies between WE and surface silicate for the analyzed period between 2002–2020. The solid black line represents the linear least-squares regression fit.

4) R2: “As another example, the authors point out a strong control of the observed changes in PFTs by light conditions in addition to nutrient limitation (l. 432-433 vs l. 435-437 and 441-442). But there is no analysis of deviations from the average light regime in the water column and how this might affect community composition. The coupled biogeochemical model used in this study is one of the few in the community, if not the only one, to have a light component (OASIM) and would therefore be a perfect tool to conduct to address such question.”

Reply: This is an important issue and we thank the reviewer for raising it. The NOBM indeed uses OASIM to simulate the propagation of downward spectral irradiance in the atmosphere and ocean, and ultimately define the level of photosynthetically available radiation (PAR) for phytoplankton growth at each depth. Due to computational constraints, the depth-resolved light field is not stored among the suite of physical and biogeochemical variables derived from the NOBM. However, we recomputed offline the light growth-limiting term experienced by phytoplankton in the mixed layer from the below ocean surface PAR estimated by OASIM and the mean mixed layer total chlorophyll concentration computed in the model. The PAR attenuation coefficient ($K_d\text{PAR}$) was empirically derived from K_d490 following Eq 9 of Morel et al. (2007) ($K_d\text{PAR} = 0.0864 + 0.884 K_d490 - 0.00137 [K_d490]^{-1}$), while K_d490 is inferred from the model mean mixed layer chlorophyll concentration following Eq 8 (Morel et al., 2007) ($K_d490 = 0.0166 + 0.0773 [\text{Chl}]^{0.6715}$). The mean mixed layer light level was retrieved using the Beer-Lambert equation, and the light limiting term was computed using the Monod-type function (Eq 6 of the Methods section in the rm) using the half-saturation irradiance (k_E) defined for each light level in the model (Table S2). We recomputed the light growth-limiting term at GOA and ENSO 3.4 for the period 2010 – 2020, which covers the onset of the anomalous warm events starting in 2013 (Figure 3 of this document). In GOA, light limitation is reduced (i.e., positive anomalies) due to shoaling of the mixed layer during the period of observed changes in PFTs composition (May–December 2014) (see also Figure S4 in the Supplementary Information). In ENSO 3.4, anomalies in the light growth-limiting term are positive given that the strong reduction in phytoplankton Chl biomass reduces light attenuation and elevates the overall light level in the mixed layer. Hence, we propose the nutrient-limiting mechanism of reduced silicate and nitrate inferred by the model as the main “bottom-up” constraint of phytoplankton growth and potential leading driver of PFTs changes in GOA and the ENSO 3.4 region during their respective anomalous warm periods. We summarize these results in lines 497–505 of the rm, within the newly added section “Model caveats and considerations”. We describe our methodology to recompute offline the mixed layer light field in the “Light limitation” subsection (line 567) of the “Methods” section in the rm.

5) R2: “The authors also argue that their results yield an alternate explanation to that of Peña et al. (2018) to explain the Chl anomaly associated with the Blob heatwave (l. 365-371). Whereas Peña et al. (2018) attributed the positive Chl anomaly to reduced grazing and/or reduced light limitation owing to increased stratification. Alternatively, the authors state that the changes in Chl result from changes in community structure associated with modification in the nutrient regime. I do not understand why the authors did not consider all the processes/parameters that are simulated by the model in order to decipher which from grazing, light, nutrient or community structure is the most important (and how they are interconnected with each other), a question that could not be addressed by, for example, Peña et al. based on field data. Again,

Figure 3: Time series of monthly anomalies in the mixed layer growth-limiting term of nitrate (blue line), silicate (green line), and light (red line) for (a) GOA and the (b) ENSO 3.4 region. Similar as in Figure 2a and 3b of the rm, red asterisks represent the periods where notorious shifts in PFTs relative composition were found in GOA (May–December 2014) and the ENSO 3.4 region (November 2015–March 2016).

I feel that the authors treated their model outputs as they would have treated "conventional" in situ data and thus did not get the most out of the modeling tool."

Reply: We propose the shift in PFT's as the main driver of higher chlorophyll concentrations during the expansion of the Blob over GOA given the observed increase in dinoflagellates and coccolithophores biomass and relative abundance with respect to that of diatoms. As discussed in section "Hints for a warmer future" of the rm, our results suggest that these alterations are atypical, as the community composition of PFTs in the North Pacific tends to be rather stable. These changes were driven by important alterations in the nutrient limiting factors, where the silica growth-limiting term, which only constrains diatom growth, showed negative anomalies during this period, while iron became increasingly available for phytoplankton with non-silica nutrient demands. While elevated surface chlorophyll concentrations were found in the past at GOA in response to increased iron deposition (Hamme et al., 2010), Peña, M. A. and Nemcek, N. and Robert, M. (2019) did not consider this hypothesis as there was no evidence of an increase in iron supply during the Blob. Our hypothesis is consistent with this observation, given that the elevated surface iron concentrations found in the model are obtained from reduced diatom drawdown and not additional supply. Light did become less limiting during this period for all groups due to increased stratification (Figure 3 above and Figure S5), but the rise in total chlorophyll as inferred by the model was not driven by an overall increase in the biomass of all

groups, but by a surge in the biomass of specific groups not limited by silica (Figure 2a, 2c, and 2e of the rm).

Enhanced stratification would tend to induce higher, and not lower, grazing rates, thereby suppressing phytoplankton chlorophyll biomass (Arteaga et al., 2020; Behrenfeld and Boss, 2018; Behrenfeld et al., 2013b), which is not consistent with what was observed at GOA during the Blob expansion. It is possible, however, that higher grazing pressure was applied unevenly between different phytoplankton groups, causing shifts in the relative abundance of PFTs. Unfortunately, under its current formulation, the NOBM is unlikely to detect or attribute changes in PFT composition due to such type of “top-down” regulation. As mentioned above, we discuss this caveat in section “Model caveats and considerations” of the rm (line 470).

6) R2: “Another important point is the range of % anomalies in the PFTs compared with the range of % errors in the model data. Although the modeled PFT data could not be validated for each of the four study regions, it seems that the range of changes (anomalies in the relative contribution of the PFTs to Chl) are lower (ex. 1. 187-196 for the GOA region; the trend is clearer for the ENSO 3.4 region) than the model error (Table S4). I think the authors should address this issue which is one of the keys of the strength of the analysis. More generally, the method used for validating the modeled PFT data is unclear. The Excel spreadsheet accessed from the URL given in p. 20 contains 300 data points for the surface or mixed layer. Some of the data appear to be cell counts, others pigment measurements. How is all this information transformed into the relative contribution of PFTs to Chl needed for model validation?”

Reply: We agree with the reviewer in that the changes in relative PFTs contribution identified at GOA are close to the model detection limits inferred from the validation exercise. The validation of the model’s PFT output is challenging due to the overall scarcity of in situ data. This is the reason why we dedicated a large portion of the discussion section of the manuscript to comparing our model-based results with studies that reported changes in phytoplankton composition due to warm anomalies based on in situ data from the Pacific Ocean (section “Ecological processes and in situ observations”, line 312 of the rm). At GOA, some of the major ecological features predicted by the model such as the decline in diatoms and increased dinoflagellates are observed in situ from Continuous Plankton Recorders (CPRs) in the northern part of the basin, as well as coastal observations collected off Newport, Oregon. These comparisons are discussed amply in lines 329–372 of the rm.

The validation data set (<https://gmao.gsfc.nasa.gov/reanalysis/MERRA-NOBM/data/phytgroups.xls>) includes 468 in situ observations from wide range of publications. We extract data on phytoplankton diversity from multiple methodologies and combine surface and mean upper mixed layer abundances in order to maximise the amount of data points available for validation. All information (either cell counts, biomass concentration (chlorophyll or carbon), and/or pigments analysis) is translated into relative abundances of phytoplankton groups (regardless of units), which is the ultimate indicator used for validation of the NOBM. The origin of the validation database can be found in Gregg et al. (2003), and has been expanded through time to include observations of newly added phytoplankton groups. A map showing the distribution of in situ observations of PFTs relative abundances (i.e., percentage of Chl-based biomass relative to total chlorophyll) used for validation of the NOBM has been added to the Supplementary Information as Figure S7.

Reviewer (R) 3

1) R3: “Review of “Impact of Pacific Ocean heatwaves on phytoplankton community composition” by L.A. Arteaga and C.S. Rousseaux

Summary of manuscript:

This study uses an assimilating global scale ocean model with a biogeochemical component to investigate the impacts of recent Pacific Ocean heatwaves on the primary phytoplankton functional types that make up the phytoplankton community. Two events are the primary focus: the north Pacific “blob” event (with a focus on the Gulf of Alaska), and the 2015–2016 El Nino with a focus on the ENSO3.4 region of the Equatorial Pacific.

Overall Impression:

The paper is well written and the findings are interesting. Even though various studies are often categorized as “timely” this one actually fits that definition since there are at least a few observationally based studies of these two events to serve as a reality check.

I have one primary reservation with the study, and also include a few minor suggestions below.

My primary hang-up regards one of the two main findings: that dinoflagellates end up accounting for the increased chlorophyll in the GOA region during the “blob” event. I question whether this is simply an artifact of the assimilation and internal model dynamics, rather than reality? The question arises because while the model assimilates satellite chlorophyll estimates, the phytoplankton functional types (PFTs) have no real check and evolve according to internal nutrient and uptake rate specifications. That approach itself seems fine, yet from my reading there is no validation/constraint that exists for dinoflagellates, whereas the other PFTs have at least some basic validation. In a sense, the dinoflagellates represent a phytoplankton error term that will increase in the absence of diatoms in order to account for the elevated satellite chlorophyll that gets assimilated. Having stated that, the dinoflagellates in the model do tend to show up under warm (stratified) conditions with lower nutrients – this is nice to see, since that is generally speaking when they tend to accumulate in the real ocean. So at least the model is aimed at producing an expected result in that regard. Checking the relevant cited observational studies, I see very little evidence to suggest that dinoflagellates accounted for the enhanced chlorophyll signal observed during the “blob” event. Perhaps the best data set comes from the Line P observations (Pena et al. 2019), but no apparent dinoflagellate increase is documented there. The Newport observations are not appropriate for this global model, as discussed by the authors. The Continuous Plankton Recorder (CPR) observations suggests a modest uptick in dinoflagellate concentration, but the cell concentrations in Batten et al. (2022) are extremely low (4,000–8,000 cells/L), even during the “blob” years. In comparison, diatom concentrations were roughly 50,000 cells/L during that period, an order of magnitude larger than the concentrations of dinoflagellates. I suppose there is an open question of how well the CPR captures either of the quantities, but the observations suggest dinoflagellate concentrations that may be within the noise of diatom concentrations. So I guess the main question is: are there any other data sources out there that indicate an explosion of dinoflagellates during the “blob” that could be pointed to in order to lend credence to this finding? At present, the referred to data is not overly convincing. If no other such data exists, I have to question whether the authors need to re-pitch the GOA

findings somehow – perhaps just as an error term? I’m honestly not sure what else to suggest but welcome the thoughts of the authors.”

Reply: Thank you for the overall positive comments on our manuscript. We agree that the lack of validation for dinoflagellates is somewhat problematic given that the increase in the surface concentration of this group signifies one of the main trade-offs in PFT composition predicted by our data-assimilation model at GOA during the expansion of the Blob. However, it is important to clarify that this does not imply that dinoflagellates serve as an error term in the model. In this version of the NOBM, the assimilation of ocean color data is constrained by chlorophyll, PIC, and aCDOM. The model computes the biomass of each PFT numerically and calculates total chlorophyll as the integrated quantity for all PFTs’ chlorophyll prior to the assimilation step. During assimilation, the insertion of satellite chlorophyll into the model field serves a bias-correction function. The final absolute concentration of each PFT might change as a function of correcting the model chlorophyll field with satellite information, but the relative composition of PFTs is preserved through the assimilation step. The validation of PFTs is conducted once the entire run is completed, and does not alter the relative abundance in PFTs predicted by the model. I.e., errors in the model estimation of diatoms and other PFTs are not minimized at the expense of the model estimation of dinoflagellates. Hence, the model would provide the same result irrespectively of whether in situ data for the validation of dinoflagellates (or any other group) are available.

We rely on the coherent climatological surface PFT and nutrient distribution patterns predicted by the model (Figure S1) as evidence of its ability to correctly map all phytoplankton groups, including dinoflagellates. As the reviewer points out, the increase in dinoflagellates is consistent with a change in environmental conditions that render the surface ocean warmer and less turbulent (Ross and Sharples, 2007). We explain the unexpected increase in surface chlorophyll during the more stratified upper ocean conditions caused by the Blob at GOA observed in situ (Peña, M. A. and Nemcek, N. and Robert, M., 2019) and in the model output with an increase in non-silica dependent phytoplankton given the reduction in the surface concentration of silica predicted by the model. The model favors dinoflagellate growth during these conditions as they have a relative high growth rate and low iron requirements compared to the other PFTs (Table S1). Unfortunately, we are not aware of other data on silica or phytoplankton diversity that could firmly confirm our model-based result. This is the reason why we dedicated a large portion of the discussion section of the manuscript to compare our model-based results with studies that reported changes in phytoplankton composition due to warm anomalies based on in situ data from the Pacific Ocean (section “Ecological processes and in situ observations”, line 312 of the rm). Nevertheless, we have revised our discussion of the CPR data to clarify that diatoms remain more abundant than dinoflagellates in terms of cell counts, despite the declining/increasing trend in diatoms/dinoflagellates observed during the warm years associated to the Blob (line 334–338 of the revised manuscript: *“In situ samples taken from Continuous Plankton Recorders (CPRs) in the northern part of the basin (53°N–58°N, 136°W–146°W) indicate a decline in diatoms and increased abundance of dinoflagellates in the warm years of 2013–2014, although the overall abundance of diatom cells remains larger than that of dinoflagellates (Batten et al., 2022).”*

2) R3: “Minor comments/suggestions:”

“Lines 32-34, awkward sentence, suggest rewording.”

Reply: Thank you for the suggestion. We have reworded this sentence to: “*High sea surface temperature (SST) anomalies continued to be recorded in the equatorial Pacific with the onset of El Niño conditions during the fall of 2014, which developed into one of the most extreme El Niño events ever recorded, reaching warm SST anomalies close to 3°C by 2015-2016 (Blunden and Arndt, 2016; Santoso et al., 2017).*”

3) R3: “Lines 77, 66, and 285 use the word notorious. Just my opinion but that’s the kind of word you should probably just use once. It makes an impact when you read it the first time, but then you roll your eyes when you read it a second time. I suggest picking one of the sentences to use the word, and then choose a different descriptor for the other sentences.”

Reply: We have edited line 65 of the revised manuscript (rm) to: “*To date, the most impactful alteration identified occurred ...*”, and line 285 to “*The largest disagreement between the anomaly in surface silicate ...*”. We have kept “notorious” only for line 76 of the rm.

4) R3: “Fig. 1 – the SST time series show assimilative model and satellite SST, and how they agree/disagree. Can a similar approach be done with the Chlorophyll signal?”

Reply: Yes, we compare the chlorophyll anomaly time series of the NOBM with that of MODIS and VIIRS for each of the four evaluated regions (GOA, ARC, NPTZ, and ENSP 3.4) in Figure S6 of the Supplementary Information (also added below as Figure 4 of this reply document).

5) R3: “Line 205, states “Negative anomalies are first observed... during the later part of 2014...” From Fig. 2e the silicate term is often negative prior to 2014, as is nitrate – e.g., from 2003–2006, but also nitrate turns negative prior to 2015. So perhaps this should be reworded?”

Reply: Thank you for detecting this. We have revised this phrase in the rm (line 204): “*During the expansion of the Blob (post 2013), negative anomalies are first observed ...*”

6) R3: “Fig. 2. Currently the nutrient anomalies in Fig. 2e,f have the same colors as the PFTs above in Fig. 2a,b, and even c,d. Even though the figure subpanels are separated, this quickly gets confusing. I suggest that the line colors in Fig. 2e,f should be changed to some other independent colors just to help minimize confusion. Ditto for Fig. 3.”

Reply: Thank you for this suggestion. We now use different/independent colors for the nutrient anomaly plots in Figure 2e,f and Figure 3e,f of the rm.

7) R3: “Fig. 3, and the discussion of the ENSO 3.4 region. Here Chlorophytes are abbreviated “Chl” and that is also used in the text for chlorophyll. These need to be differentiated better.”

Figure 4: Time series of monthly surface chlorophyll anomalies from the NOBM (blue line), MODIS (red line) and VIIRS (green line), in the four evaluated regions: (a) GOA, (b) ARC, (c) NPTZ, and (d) ENSO 3.4.

Reply: Thank you for highlighting this potential source of confusion. We have changed the abbreviation for chlorophytes from “Chl” to “Chpt” in Figure 2, Figure 3, and Figure 7 of the rm.

8) R3: “Line 269, suggest deleting “towards higher depths””

Reply: We have removed “towards higher depths” from line 269 of the rm.

9) R3: “Fig. 4. Panels c, and d have axes labels that are way too small.”

Reply: We have enlarged the labels of all panels in Figure 4 of the rm.

10) R3: “Line 315, regarding iron and nitrate requirements... is this simply because of the division rate being different (the k values in Table S1 for diatoms and dinos are the same)?”

Reply: Many thanks for pointing out this inconsistency. We have corrected this in line 315 of the rm to indicate that *“In the biogeochemical model, the nitrate and iron requirements of dinoflagellates are the same as for diatoms (Table S1).”*

11) R3: “Fig. 7 right side panels, you might consider an alternative color for the dinoflagellates since magenta is difficult to see alongside red.”

Reply: We have changed the dinoflagellates’ marker to dark green in Figure 7 of the rm. Dinoflagellates are mostly observed in the upper right corner of the GOA panel (high-SST and high-Chl anomalies).

12) R3: “Line 539, “bugget”, do you mean “budget””

Reply: Thank you for spotting this typo. “Budget” is corrected in line 600 of the rm.

13) R3: “Regarding model validation. I don’t find the chlorophyll “validation” all that useful, particularly a global annual average statistic. What does it even mean to validate an assimilative model? I suppose it just tries to quantify how well the model keeps what you gave it?”

Reply: Yes. The aim of the NOBM is to represent as close as possible the ocean color fields inferred from satellite observations that are validated against in situ observations (such as chlorophyll), while covering gaps and correcting the remote sensing retrievals using mechanistic biogeochemical and ocean circulation dynamics. The assimilation exercise corrects not only the model, but also corrects sampling errors in the satellite data. Ocean color sensors only observe about 15 % of the global oceans per day, while models can provide complete daily coverage. The 15 % of the satellite observed oceans occur in the best places and times for phytoplankton growth: the highest solar elevations and the clearest skies. Persistent clouds can obscure some regions to three or fewer observations per month. In the high latitudes, whole seasons are missing. This can lead to important biases in satellite-based estimates of chlorophyll a (Gregg and Casey, 2007b).

14) R3: "Line 562, the link to climatological data needs updating"

Reply: The link has been updated in line 631 of the rm (please copy and paste the link into a browser as the automatic redirecting of the pdf can fail): <https://gmao.gsfc.nasa.gov/reanalysis/MERRA-NOBM/data/phytgroups.xls>

15) R3: "Line 564, the PF groups in the Excel spreadsheet I found in relation to the bad link provided do not appear to have any months associated with them. So where is that information for evaluating the model? Perhaps I landed on the wrong spreadsheet?"

Reply: Month information is in the first two columns of the spreadsheet linked above.

References

- Arteaga, L. A., Boss, E., Behrenfeld, M. J., Westberry, T. K., and Sarmiento, J. L. (2020). Seasonal modulation of phytoplankton biomass in the Southern Ocean. *Nature Communications*, 11:5364.
- Barth, A., Walter, R. K., Robbins, I., and Pasulka, A. (2020). Seasonal and interannual variability of phytoplankton abundance and community composition on the Central Coast of California. *Marine Ecology Progress Series*, 637:29–43.
- Batten, S. D., Ostle, C., Hélaouët, P., and Walne, A. W. (2022). Responses of Gulf of Alaska plankton communities to a marine heat wave. *Deep Sea Research Part II: Topical Studies in Oceanography*, 195:105002.
- Behrenfeld, M. J. and Boss, E. S. (2018). Student’s tutorial on bloom hypotheses in the context of phytoplankton annual cycles. *Global Change Biology*, 24(1):55–77.
- Behrenfeld, M. J., Doney, S. C., Lima, I., Boss, E. S., and Siegel, D. A. (2013a). Annual cycles of ecological disturbance and recovery underlying the subarctic Atlantic spring plankton bloom. *Global Biogeochemical Cycles*, 27(2):526–540.
- Behrenfeld, M. J., Hu, Y., Hostetler, C. A., Dall’Olmo, G., Rodier, S. D., Hair, J. W., and Trepte, C. R. (2013b). Space-based lidar measurements of global ocean carbon stocks. *Geophysical Research Letters*, 40(16):4355–4360.
- Blunden, J. and Arndt, D. S. (2016). State of the Climate in 2015. *Bulletin of the American Meteorological Society*, 97(8):s1–s275.
- Bond, N. A., Cronin, M. F., Freeland, H., and Mantua, N. (2015). Causes and impacts of the 2014 warm anomaly in the NE Pacific. *Geophysical Research Letters*, 42(9):3414–3420.
- Di Lorenzo, E. and Mantua, N. (2016). Multi-year persistence of the 2014/15 North Pacific marine heatwave. *Nature Climate Change*, 6:1042–1047.
- Du, X. and Peterson, W. T. (2018). Phytoplankton community structure in 2011–2013 compared to the extratropical warming event of 2014–2015. *Geophysical Research Letters*, 45(3):1534–1540.
- Gelaro, R., McCarty, W., Suárez, M. J., Todling, R., Molod, A., Takacs, L., Randles, C. A., Darmenov, A., Bosilovich, M. G., Reichle, R., Wargan, K., Coy, L., Cullather, R., Draper, C., Akella, S., Buchard, V., Conaty, A., da Silva, A. M., Gu, W., Kim, G.-K., Koster, R., Lucchesi, R., Merkova, D., Nielsen, J. E., Partyka, G., Pawson, S., Putman, W., Rienecker, M., Schubert, S. D., Sienkiewicz, M., and Zhao, B. (2017). The modern-era retrospective analysis for research and applications, version 2 (merra-2). *Journal of Climate*, 30(14):5419 – 5454.
- Gregg, W. W. and Casey, N. W. (2007a). Modeling coccolithophores in the global oceans. *Deep Sea Research Part II: Topical Studies in Oceanography*, 54(5):447–477. The Role of Marine Organic Carbon and Calcite Fluxes in Driving Global Climate Change, Past and Future.
- Gregg, W. W. and Casey, N. W. (2007b). Sampling biases in modis and seawifs ocean chlorophyll data. *Remote Sensing of Environment*, 111(1):25–35.

- Gregg, W. W., Casey, N. W., and Rousseaux, C. S. (2013). Global surface ocean carbon estimates in a model forced by MERRA. NASA Technical Report Series on Global Modeling and Data Assimilation, NASA TM-2013-104606. Technical report, NASA.
- Gregg, W. W., Ginoux, P., Schopf, P. S., and Casey, N. W. (2003). Phytoplankton and iron: validation of a global three-dimensional ocean biogeochemical model. *Deep Sea Research Part II: Topical Studies in Oceanography*, 50(22):3143–3169. The US JGOFS Synthesis and Modeling Project: Phase II.
- Hamme, R. C., Webley, P. W., Crawford, W. R., Whitney, F. A., DeGrandpre, M. D., Emerson, S. R., Eriksen, C. C., Giesbrecht, K. E., Gower, J. F. R., Kavanaugh, M. T., Peña, M. A., Sabine, C. L., Batten, S. D., Coogan, L. A., Grundle, D. S., and Lockwood, D. (2010). Volcanic ash fuels anomalous plankton bloom in subarctic northeast Pacific. *Geophysical Research Letters*, 37(19).
- Morel, A., Huot, Y., Gentili, B., Werdell, P. J., Hooker, S. B., and Franz, B. A. (2007). Examining the consistency of products derived from various ocean color sensors in open ocean (case 1) waters in the perspective of a multi-sensor approach. *Remote Sensing of Environment*, 111(1):69 – 88.
- Peña, M. A. and Nemcek, N. and Robert, M. (2019). Phytoplankton responses to the 2014–2016 warming anomaly in the northeast subarctic Pacific Ocean. *Limnology and Oceanography*, 64(2):515–525.
- Ross, O. and Sharples, J. (2007). Phytoplankton motility and the competition for nutrients in the thermocline. *Marine Ecology Progress Series*, 347:21–38.
- Santoso, A., Mcphaden, M. J., and Cai, W. (2017). The Defining Characteristics of ENSO Extremes and the Strong 2015/2016 El Niño. *Reviews of Geophysics*, 55(4):1079–1129.
- Whitney, F. A. (2015). Anomalous winter winds decrease 2014 transition zone productivity in the NE Pacific. *Geophysical Research Letters*, 42(2):428–431.

REVIEWERS' COMMENTS:

Reviewer #1 (Remarks to the Author):

I would like to thank the authors of the manuscript for their careful revisions. I'm pleased with the revised manuscript and I'm happy to recommend it for publication.

Reviewer #2 (Remarks to the Author):

The authors have satisfactorily addressed most of my comments. I therefore recommend the publication of this manuscript.

Reviewer #3 (Remarks to the Author):

The authors have addressed my minor comments, and I thank them for that. I think the altered colors for lines helps the reader to more easily distinguish the different variables.

My primary concern was the explosion of dinoflagellates in the GOA region of the model, and the lack of any evidence of such a transformation in the real ocean. I appreciate the authors' explanation that the model assimilation does not alter the relative percentage of the PFTs. Unfortunately, it sounds like the authors are not aware of any other data that might be available to help address the reality of this result. The only applicable data come from Peña et al. (2019) and Batten et al. (2022). No increase in dinoflagellates was documented by Peña et al. (2019). The Batten et al. (2022) data show that diatom cell concentrations in 2014 were roughly 30,000 cells/L whereas dinoflagellates were only 3,000 cells/L. In 2015, those same two groups were 50,000 cells/L and 4,000 cells/L. In other words, according to the observations presented in Batten et al. (2022), dinoflagellate concentrations were exceedingly low and were, at most, 10% of the diatom concentrations in the GOA.

The authors have been very clear and careful in their writing. However, since no observational evidence exists to support the dinoflagellate bloom in the GOA region, I suggest that the authors recast this particular result as a 'hypothesis' for what may have happened in the real GOA during the blob period. Given the lack of observational data, the fact that errors in the model silicate field itself can be on the order of ~50% (which could impact diatom vs. dinoflagellate growth in the model), this seems to be best way forward. Assuming the authors are doubly clear about that, then I see no issue with publishing the paper. I do not think these changes will take more than a couple or three targeted sentences and so consider those to be very minor changes.